# Refined Mechanism Design for Approximately Structured Priors via Active Regression

**Christos Boutsikas**
Purdue University
cboutsik@purdue.edu

**Petros Drineas**
Purdue University
pdrineas@purdue.edu

**Marios Mertzanidis**
Purdue University
mmertzan@purdue.edu

**Alexandros Psomas**
Purdue University
apsomas@cs.purdue.edu

**Paritosh Verma**
Purdue University
verma136@purdue.edu

## Abstract

We consider the problem of a revenue-maximizing seller with a large number of items $m$ for sale to $n$ strategic bidders, whose valuations are drawn independently from high-dimensional, unknown prior distributions. It is well-known that optimal and even approximately-optimal mechanisms for this setting are notoriously difficult to characterize or compute, and, even when they can be found, are often rife with various counter-intuitive properties. In this paper, following a model introduced recently by Cai and Daskalakis [CD22], we consider the case that bidders' prior distributions can be well-approximated by a topic model. We design an active learning component, responsible for interacting with the bidders and outputting low-dimensional approximations of their types, and a mechanism design component, responsible for robustifying mechanisms for the low-dimensional model to work for the approximate types of the former component. On the active learning front, we cast our problem in the framework of Randomized Linear Algebra (RLA) for regression problems, allowing us to import several breakthrough results from that line of research, and adapt them to our setting. On the mechanism design front, we remove many restrictive assumptions of prior work on the type of access needed to the underlying distributions and the associated mechanisms. To the best of our knowledge, our work is the first to formulate connections between mechanism design, and RLA for active learning of regression problems, opening the door for further applications of randomized linear algebra primitives to mechanism design.

## 1 Introduction

The design of revenue-optimal auctions is a central problem in Economics and Computer Science. In this problem, a revenue-maximizing seller has $m$ heterogeneous items for sale to $n$ strategic bidders. Each bidder $i$ has a type $\mathbf{t}_i \in \mathbb{R}^d$ which contains enough information to encode the bidder's willingness to pay for every subset of items. Bidders' types are private information, and thus, in order to provide meaningful guarantees on the seller's revenue, the standard approach in Economics is to make a Bayesian assumption: types are drawn from a joint distribution $\mathcal{D}$.

Assuming a single item for sale and bidders' types that are drawn independently from known distributions, Myerson's seminal work [Mye81] provides a closed-form solution for the revenue-optimal mechanism. Beyond this single-item case, however, multi-item mechanism design remains an active research agenda, even forty years later. Optimal mechanisms are no longer tractable, in any sense of the word, as well as exhibit various counter-intuitive properties [MV07, DDT13, DDT15, BCKW15, HR15, HN19, PSCW22]; see [Das15] for a survey. On the other

37th Conference on Neural Information Processing Systems (NeurIPS 2023).

hand, significant research effort has culminated in numerous compelling positive results, such as simple and approximately optimal auctions [CHK07, CHMS10, CMS15, Yao15, RW15, CM16, CDW16, CZ17, KW19, BILW20], as well as efficient algorithms for computing near-optimal auctions [AFH+12, CDW12a, CDW12b, CDW13a, CDW13b], even with just sampling access to the type distribution [CR14, HMR15, DHP16, MR16, CD17, GHZ19, BCD20, GW21].

Despite all this progress, however, there are key challenges in applying these results in practice. First, the computational complexity, sample complexity, approximation guarantees, and communication complexity (i.e., the amount of information the bidder should communicate to the seller) of these results often depend on the number of items $m$, which could be prohibitively large (e.g., think of $m$ as the number of items on `Amazon.com`). Second, bidders' type distributions are typically high-dimensional, or otherwise complex objects, that are not known nor can they be sampled. Instead, the designer might have an estimated distribution $\hat{\mathcal{D}}$, e.g., through market research, that is close to the real distribution $\mathcal{D}$. Motivated by these issues, Cai and Daskalakis [CD22] introduce a model where the true type distribution $\mathcal{D}_i$ of bidder $i$ is close to a structured distribution $\hat{\mathcal{D}}_i$. Specifically, they assume that there is an underlying design matrix $\mathbf{A} \in \mathbb{R}^{m \times k}$ of $k$ "archetypes," with $k \ll m$. Intuitively, bidder $i$ can be approximated by a linear combination of $k$ archetypal bidders. This same assumption has been central in the study of recommender systems. In this model, [CD22] give a framework for transforming a mechanism $\hat{\mathcal{M}}$ for the low-dimensional distribution $\hat{\mathcal{D}}_z$ into a mechanism for the true type distribution with good revenue guarantees, and whose computational and communication complexity does not depend on the number of items $m$.

The impact of their work notwithstanding, the above results require strong structural assumptions on: the design matrix $\mathbf{A}$; the bidders' valuation functions; and very specific access to (or exact knowledge of) the structured distribution $\hat{\mathcal{D}}_z$ and the mechanism $\hat{\mathcal{M}}$. *Our work connects the recommender system approaches for mechanism design with recent progress on Randomized Linear Algebra for active learning for regression problems. We relax, and even remove these restrictive assumptions, and open the door to future exploration of more elaborate recommender system models in the context of mechanism design, using randomized linear algebra primitives.*

**The framework and results [CD22].** To place our results in context, we start with a brief overview of [CD22], which considers a setting where the type distribution $\mathcal{D}_i$ of bidder $i$ is close to a distribution $\hat{\mathcal{D}}_i$ in the Prokhorov distance, under the $\ell_\infty$ norm (Definition 1). Here, $\hat{\mathcal{D}}_i$ first samples a vector $\mathbf{z} \in [0, 1]^k$ from a low-dimensional distribution $\hat{\mathcal{D}}_{z,i}$, and then outputs $\mathbf{A}\mathbf{z}$, where $\mathbf{A} \in \mathbb{R}^{m \times k}$ is a known matrix. The proposed framework has a learning and a mechanism design component.

Their learning component consists of a communication-efficient[1] query protocol $\mathcal{Q}$ for interacting with each bidder $i$ such that, if the type $\mathbf{t}_i$ of bidder $i$ satisfies $\|\mathbf{t}_i - \mathbf{A}\mathbf{z}\|_\infty \leq \varepsilon$, the query protocol outputs a vector $\mathcal{Q}(\mathbf{t}_i)$ such that $\|\mathcal{Q}(\mathbf{t}_i) - \mathbf{z}\|_\infty \leq \zeta_\infty$.[2] [CD22] give such query protocols under strong (distinct) conditions on $\mathbf{A}$, and specifically when $\mathbf{A}$: *(i)* satisfies an assumption similar to the separability condition of [DS03], *(ii)* is generated from a distribution where each archetype is an independent copy of an $m$-dimensional Gaussian, or *(iii)* is generated from a distribution where each archetype is an independent copy of an $m$-dimensional, bounded distribution with weak dependence. We discuss these restrictions in more detail in Appendix D.4.1. The query complexity, as well as the error $\zeta_\infty$, depend on them, but, importantly, are independent of the number of items $m$.

Their mechanism design component is a refinement of a robustification result of [BCD20]. For this transformation to work, one needs to interact with the mechanism and the underlying distributions using highly non-trivial operations, which are computationally demanding, and require exact knowledge of bidders' valuation functions. In our work we overcome this issues by developing new reductions and plugging them in the framework established by [BCD20] and [CD22]. The overall interplay between the different mechanism design and active regression components can be seen in Fig. 1.

Combining the two components, [CD22] obtain mechanisms for $\mathcal{D}$, for constrained-additive bidders.[3] In these mechanisms, each bidder is required to answer a small number of simple `Yes/No` queries

---

[1]By efficient we mean a query protocol that asks each bidder a small amount of queries. Se also Definition 3.

[2]Recall that for any (integer) $1 \leq p < \infty$ and a vector $\mathbf{x} \in \mathbb{R}^d$, $\|\mathbf{x}\|_p^p = \sum_{i=1}^d |\mathbf{x}_i|^p$; for $p = \infty$, $\|\mathbf{x}\|_\infty = \max_{i=1...d} |\mathbf{x}_i|$. See [CD22] for an exact expression for $\zeta_\infty$.

[3]A function is constrained-additive if $v(t, S) = \max_{T \in \mathcal{I} \cap 2^S} \sum_{j \in T} t_j$, where $\mathcal{I}$ is a downward-closed set system.

of the form "are you willing to pay $p$ for item $j$?", such that the loss in revenue and the violation in incentive compatibility do not depend on the number of items $m$.

## 1.1 Our contributions

**Randomized Linear Algebra (RLA) for active learning.** RLA for active learning has focused on solving regression problems of the form

$$\mathbf{z} = \arg\min_{\mathbf{x}} \|\mathbf{t} - \mathbf{A}\mathbf{x}\|_p,$$

for $\ell_p$ norms with $1 \leq p < \infty$, by querying *only* a subset of elements of the vector $\mathbf{t}$. In prior work on RLA for active learning, the focus has been on recovering an approximate solution that achieves a relative error or constant factor approximation to the optimum loss. We adapt these bounds to our setting, which could include noisy instead of exact queries, and prove bounds for the $\ell_p$ norm error of the exact minus the approximate solution. Specifically, we bound $\|\mathcal{Q}(\mathbf{t}_i) - \mathbf{z}\|_p \leq \zeta_p$. We provide bounds on $\zeta_p$ for all integers $1 \leq p < \infty$. Our bounds depend on the modelling error $\|\mathbf{t} - \mathbf{A}\mathbf{x}\|_p$ and some measure of the query noise (see Definitions 4 and 5); both dependencies are expected. Importantly, our bounds hold for *arbitrary archetype matrices*, very much unlike the work of [CD22], which focused on very restricted classes of matrices. A single property of the archetype matrix, the smallest singular value with respect to the $\ell_p$ induced matrix norm, $\sigma_{\min,p}(\mathbf{A})$, can characterize the quality of the error. As $\sigma_{\min,p}(\mathbf{A})$ decreases, the error $\zeta_p$ grows. Intuitively, $\sigma_{\min,p}(\mathbf{A})$ is a measure of independence of the archetypes, with small values corresponding to linearly dependent archetypes. The query complexity needed to achieve error $\zeta_p$ is almost linear (up to logarithmic factors) on $k$ for $p = 1$ and $p = 2$, and grows with $k^{p/2}$ for $p \geq 3$. Our query complexity bounds have no dependency on $m$ (number of items) or $d$ (dimensionality of a type) enabling us to produce results way beyond constrained-additive valuations, as $d = 2^m$ dimensions suffice to encode *arbitrary* valuation functions.

It is critical to highlight that our ability to provide bounds on the approximation error for arbitrary archetype matrices is, at least partly, due to leveraging information from the archetype matrix $\mathbf{A}$. Specifically, we use this information to select which bidder types to query, instead of just querying types uniformly at random. This information involves the computation or approximation of the well-studied leverage scores of $\mathbf{A}$ for $p = 2$ and of the so-called Lewis weights for all other values of $p$ (see Section 3.1). We do note that in our framework, the errors due to the modeling of the bidder type $\mathbf{t}$ as the product $\mathbf{A}\mathbf{z}$ and the query noise are always bounded by the respective $\ell_p$ norm. Thus, our models are more restrictive compared to the $\ell_\infty$ norm models of [CD22]. However, to the best of our knowledge, even assuming such restrictive models, the results and tools of prior work do not extend to arbitrary archetype matrices. This is precisely the gap that is bridged by our work, using RLA for active learning of $\ell_p$ norm regression for $1 \leq p < \infty$.

**Mechanism Design.** On the mechanism design front, our main contribution is relaxing the assumptions of [CD22, BCD20] on the type of access needed to the low-dimensional distribution and the mechanism for it. Specifically, we further refine the robustification result of Brustle et al. [BCD20] and remove the need for the aforementioned strong oracle.

The main difficulty of transforming mechanisms for one distribution into mechanisms for another distribution is that the two distributions might not share the same support. The crux of the issue is that the incentive constraints are very delicate; a small change in the underlying distribution may drastically change the agents' *valuation distribution* over the mechanisms' outcomes. One way to tame the distribution of outcomes is to map bids that are not in the support of the initial distribution, to bids that are. Brustle et al. [BCD20] do this by "optimally misreporting" on behalf of the bidder, by calculating $\arg\max_{\mathbf{t}'_i \in supp(\widetilde{\mathcal{D}}_{z,i})} \mathbb{E}_{\mathbf{b}_{-i} \sim \widetilde{\mathcal{D}}_{z,-i}}[u_i(v_i, \hat{\mathcal{M}}(\mathbf{t}'_i, \mathbf{b}_{-i}))]$, where $\widetilde{\mathcal{D}}_{z,i}$ is a rounded-down version of $\hat{\mathcal{D}}_{z,i}$. As we've discussed, for this operation to be viable, many things need to be assumed about what the designer knows and can compute about $\hat{\mathcal{M}}, \hat{\mathcal{D}}_{z,i}$, and the bidder's valuation function. Our approach, instead, leverages the fact that when two distributions are close in Prokhorov distance, under any $\ell_p$ norm, *any* point on the support of one distribution is close to a point on the support of the other, with high probability. Our construction simply maps a report $\mathbf{w}_i$ to the "valid" report (approximately) closest to $\mathbf{w}_i$ in $\ell_p$ distance. This operation is linear on the support size. Furthermore, our overall robustification result holds for all norms, not just $\ell_\infty$, and our construction is completely agnostic to bidders' valuation functions.

**Combining the components.** Combining the two components we can, without any assumptions on **A**, given a mechanism for the low-dimensional prior, design mechanisms with comparable revenue guarantees, where each bidder is required to answer a small number of queries. Our queries ask a bidder her value for a subset of items, and our mechanism can accommodate *any* valuation function, significantly extending the scope of our results.

**Related Work.** Aside from the work of [CD22], on the mechanism design front, the most relevant works are [CD17], that consider learning multi-item auctions given "approximate" distributions, and [BCD20], that consider learning multi-item auctions when type distributions are correlated, yet admit special structure. On the RLA front, we leverage and adapt multiple recent results on approximating $\ell_p$ regression problems in an active learning setting. We discuss prior work on RLA for active learning and its connections to our setting in Section 3.1 and Appendix D.

## 2 Preliminaries

Let $[n] \coloneqq \{1, 2, \ldots, n\}$ denote the first $n$ natural numbers. A revenue-maximizing seller has a set $[m]$ of $m$ heterogeneous items for sale to $n$ strategic bidders indexed by $[n]$. Bidder $i$ has a private type vector $\mathbf{t}_i \in \mathbb{R}^d$, and the types of all bidders are represented by a *valuation profile* $\mathbf{t} = (\mathbf{t}_1, \ldots, \mathbf{t}_n)$. Bidder $i$ has a valuation function $v_i : \mathbb{R}^d \times 2^{[m]} \to \mathbb{R}_+$, that takes as input the bidder's type and a (possibly randomized[4]) set of items $S \subseteq [m]$ and outputs the bidder's value for $S$. Note that $d \leq 2^m$, since expressing a valuation function requires at most one real number per subset of items. Types are drawn independently. Let $\mathcal{D} = \times_{i \in [n]} \mathcal{D}_i$ be the distribution over bidders' types, $\mathcal{D}_{-i} = \times_{j \in [n]/\{i\}} \mathcal{D}_j$ be the distribution of all bidders excluding $i$, and $supp(\mathcal{D})$ be the support of a distribution $\mathcal{D}$.

**Mechanisms.** A mechanism $\mathcal{M} = (x, p)$ is a tuple where $x : \mathbb{R}_+^{nd} \to 2^{[nm]}$ is the allocation rule, and $p : \mathbb{R}_+^{nd} \to \mathbb{R}_+^n$ is the payment rule, which map *reported* types to allocations of the items and payments, respectively. Specifically, $x_{i,j}(\mathbf{b}) \coloneqq (x(\mathbf{b}))_{i,j}$ denotes the probability that bidder $i$ receives item $j$ for input valuation profile $\mathbf{b}$, and $p_i(\mathbf{b}) \coloneqq (p(\mathbf{b}))_i$ denotes the amount bidder $i$ has to pay. Let $u_i(\mathbf{t}_i, \mathcal{M}(\mathbf{b}))$ be the utility of bidder $i$ with type $\mathbf{t}_i$ for participating in mechanism $\mathcal{M}$, under reports $\mathbf{b}$. Bidders are risk-free and quasi-linear i.e., $u_i(\mathbf{t}_i, \mathcal{M}(\mathbf{b})) = \mathbb{E}\left[v_i(\mathbf{t}_i, x(\mathbf{b})) - p_i(\mathbf{b})\right]$, where the expectation is taken over the randomness of the allocation rule. Since we only consider truthful mechanisms, unless stated otherwise, reported types will be the same as the true types.

A bidder's objective is to maximize her utility. The seller strives to design mechanisms that incentivize bidders to report truthfully. We use the following notions of truthfulness. A mechanism $\mathcal{M}$ is $\varepsilon$-*Bayesian Incentive Compatible ($\varepsilon$-BIC)*, if for each bidder $i \in [n]$, any type $\mathbf{t}_i$ and misreport $\mathbf{t}_i'$ we have that: $\mathbb{E}_{\mathbf{t}_{-i} \sim \mathcal{D}_{-i}}[u_i(\mathbf{t}_i, \mathcal{M}(\mathbf{t}_i, \mathbf{t}_{-i}))] \geq \mathbb{E}_{\mathbf{t}_{-i} \sim \mathcal{D}_{-i}}[u_i(\mathbf{t}_i, \mathcal{M}(\mathbf{t}_i', \mathbf{t}_{-i}))] - \varepsilon$. A mechanism $\mathcal{M}$ is $(\varepsilon, \delta)$-*BIC* if for each bidder $i \in [n]$, and any misreport $\mathbf{t}_i'$ we have that:

$$\mathbb{P}_{\mathbf{t}_i \sim \mathcal{D}_i}\left[\mathbb{E}_{\mathbf{t}_{-i} \sim \mathcal{D}_{-i}}[u_i(\mathbf{t}_i, \mathcal{M}(\mathbf{t}_i, \mathbf{t}_{-i}))] \geq \mathbb{E}_{\mathbf{t}_{-i} \sim \mathcal{D}_{-i}}[u_i(\mathbf{t}_i, \mathcal{M}(\mathbf{t}_i', \mathbf{t}_{-i}))] - \varepsilon\right] \geq 1 - \delta.$$

A $(\varepsilon, 0)$-BIC mechanism is a $\varepsilon$-BIC mechanism; a 0-BIC mechanism is simply BIC. Finally, a mechanism $\mathcal{M}$ is *ex-post Individually Rational (IR)* if for all valuation profiles $\mathbf{t}$ and all bidders $i \in n$, $u_i(\mathbf{t}_i, \mathcal{M}(\mathbf{t}_i, \mathbf{t}_{-i})) \geq 0$. The seller's objective is to maximize her expected *revenue*. For a mechanism $\mathcal{M}$ and distribution $\mathcal{D}$ we denote the expected revenue as $Rev(\mathcal{M}, \mathcal{D}) = \mathbb{E}_{\mathbf{t} \sim \mathcal{D}}\left[\sum_{i \in [n]} p_i(\mathbf{t})\right]$. Note that, we are calculating revenue assuming truthful reports, even for, e.g., $(\varepsilon, \delta)$-BIC mechanisms.

**Statistical Distance.** In this work we design mechanisms that work well, as long as they are evaluated on distributions that are "close" to $\mathcal{D}$. Here, we define the notion of distance between two probability measures that we use throughout the paper.

**Definition 1** (Prokhorov Distance). *Let $(\mathcal{U}, d)$ be a metric space and $\mathcal{B}$ be a $\sigma$-algebra on $\mathcal{U}$. For $A \in \mathcal{B}$, let $A^\varepsilon = \{x : \exists y \in A \text{ s.t. } \pi(x, y) < \varepsilon\}$ where $\pi$ is some distance metric. Two probability measures $P$, $Q$ on $\mathcal{B}$ have Prokhorov distance: $\inf\{\varepsilon > 0 : P(A) \leq Q(A^\varepsilon) + \varepsilon \text{ and } Q(A) \leq P(A^\varepsilon) + \varepsilon, \forall A \in \mathcal{B}\}$. We choose $\pi$ to be the $\ell_p$-distance, and we denote the Prokhorov distance between measures $P, Q$ as $\pi_p(P, Q)$.*

---

[4]Each subset might be selected according to a probability distribution.

The following is an equivalent characterization of Prokhorov distance due to Strassen [Str65].

**Lemma 1** ([Str65]). *Let $\mathcal{D}$ and $\mathcal{D}'$ be two distributions supported on $\mathbb{R}^n$. $\pi_p(\mathcal{D}, \mathcal{D}') \leq \varepsilon$ iff there exists coupling $\gamma$ of $\mathcal{D}$ and $\mathcal{D}'$, such that $\mathbb{P}_{(\mathbf{x},\mathbf{y}) \sim \gamma}[\|\mathbf{x} - \mathbf{y}\|_p > \varepsilon] \leq \varepsilon$.*

**Recommendation system-inspired model.** We assume that, for each bidder, there exists a known design matrix $\mathbf{A} \in \mathbb{R}^{d \times k}$ [5], where the columns of $\mathbf{A}$ represent $k$ "archetypes", for a constant $k$. Our results hold if these matrices are different for each bidder, however, for ease of notation we will assume all bidders have the same design matrix. For each bidder $i$ there exists a distribution $\hat{\mathcal{D}}_{z,i}$ supported on the latent space $[0,1]^k$. Let $\hat{\mathcal{D}}_i = \mathbf{A} \circ \hat{\mathcal{D}}_{z,i}$ be the distribution induced by multiplying a sample from $\hat{\mathcal{D}}_{z,i}$ with the design matrix $\mathbf{A}$, i.e., $\hat{\mathcal{D}}_i$ is the distribution of $\mathbf{A}\mathbf{y}$ where $\mathbf{y} \sim \hat{\mathcal{D}}_{z,i}$.

The valuation function over the latent types is defined as $v_i^{\mathbf{A}}(\mathbf{z}_i, S) := v_i(\mathbf{A}\mathbf{z}_i, S)$ for any bundle $S \subseteq [m]$. We will use the following notion of Lipschitz continuity for valuation functions.

**Definition 2** (Lipschitz Valuation). *A valuation function $v(\cdot, \cdot) : \mathbb{R}^d \times [0,1]^m \to \mathbb{R}_+$ is $\mathcal{L}$-Lipschitz, if for any two types $\mathbf{t}, \mathbf{t}' \in \mathbb{R}^d$ and any bundle $S \subseteq [m]$, $|v(\mathbf{t}, S) - v(\mathbf{t}', S)| \leq \mathcal{L}\|\mathbf{t} - \mathbf{t}'\|_\infty$.*

We include a table, Table 1, with all the notation used throughout the paper in the appendix.

# 3 Active Learning for Regression and Mechanism Design

In this section, we state our main results, deferring all technical proofs to the appendix. We present mechanisms that are completely agnostic with respect to $\mathcal{D}$, the distribution from which bidders' types are drawn from. However, we have limited access (described later in this section) to *(i)* a design matrix $\mathbf{A}$; *(ii)* distributions $\hat{\mathcal{D}}_{z,i}$ over $\mathbb{R}^k$, where for all $i \in [n]$, $\pi_p(\mathcal{D}_i, \mathbf{A} \circ \hat{\mathcal{D}}_{z,i}) \leq \varepsilon_{\mathtt{mdl},p}$ for some $\varepsilon_{\mathtt{mdl},p} > 0$; and *(iii)* a mechanism $\hat{\mathcal{M}}$ for $\hat{\mathcal{D}}_z = \times_{i \in [n]}\hat{\mathcal{D}}_{z,i}$. This limited access to the design matrix motivates the use of active learning, which deals precisely with settings where the algorithm is allowed to (interactively) query a subset of the available data points for their respective labels (see [MD21, MMWY22] for precise definitions of the active learning setting in regression problems). Our approach is modular and starts by building an active learning component for regression problems (Section 3.1) followed by the mechanism design component (Section 3.2). We combine the two components to get an overall mechanism for $\mathcal{D}$ in Section 3.3.

## 3.1 Active learning for regression via Randomized Linear Algebra

Our objective is to design a communication-efficient, active learning query protocol for the seller that interacts with each bidder $i$, and infers their type $\mathbf{t}_i \in \mathbb{R}^d$ by accessing only a small subset of elements of the type vector (as $d$ is very large). We use $\mathcal{Q}$ to denote the query protocol, whose output is a vector in the low-dimensional latent space $\mathbb{R}^k$. A bidder interacts with the query protocol *truthfully* if it is in her best interest to evaluate functions requested by the protocol on her true private type $\mathbf{t}_i$. We use $\mathcal{Q}(\mathbf{t}_i) \in \mathbb{R}^k$ to denote the output of $\mathcal{Q}$ when interacting with a truthful bidder with type $\mathbf{t}_i$. We now define the notion of an $(\varepsilon_{\mathtt{mdl},p}, \zeta_p, p)$-query protocol and the notion of *query noise*.

**Definition 3** (($\varepsilon_{\mathtt{mdl},p}, \zeta_p, p$)-query protocol). *$\mathcal{Q}$ is called an $(\varepsilon_{\mathtt{mdl},p}, \zeta_p, p)$-query protocol, if, for all $\mathbf{t} \in \mathbb{R}^d$ and $\mathbf{z} \in \mathbb{R}^k$ satisfying $\|\mathbf{t} - \mathbf{A}\mathbf{z}\|_p \leq \varepsilon_{\mathtt{mdl},p}$, we have $\|\mathbf{z} - \mathcal{Q}(\mathbf{t})\|_p \leq \zeta_p$.*

**Definition 4** (Query noise). *Let $\mathbf{t}_i$ be the true type of a bidder. Our query protocol can access entries of $\mathbf{t}_i + \boldsymbol{\epsilon}_{\mathtt{nq},p}$, where $\boldsymbol{\epsilon}_{\mathtt{nq},p}$ is an (unknown) vector. The query noise $\varepsilon_{\mathtt{nq},p}$ satisfies $\|\boldsymbol{\epsilon}_{\mathtt{nq},p}\|_p \leq \varepsilon_{\mathtt{nq},p}$.*

The query noise depends on the specifics of the interactions between the seller and the bidder. For example, if the seller is only allowed to ask queries of the form "'what is your value for the subset $S$?", the query noise $\varepsilon_{\mathtt{nq},p}$ is equal to zero. Our bounds will also depend on the *model error*.

**Definition 5** (Model error). *Given a valuation profile $\mathbf{t} \in \mathbb{R}^{nd}$, the* model error *is $\varepsilon_{\mathtt{mdl},p}$ if, for all $i \in [n]$, there exists a $\mathbf{z}_i \in \mathbb{R}^k$ such that $\|\mathbf{t}_i - \mathbf{A}\mathbf{z}_i\|_p \leq \varepsilon_{\mathtt{mdl},p}$.*

Note that, we don't have bounds of the form "$\|\mathbf{t}_i - \mathbf{A}\mathbf{z}_i\| \leq \varepsilon$" for individual types, but for the distributions $\mathcal{D}_i$ and $\mathbf{A} \circ \hat{\mathcal{D}}_{z,i}$. The characterization of Prokhorov distance (Lemma 1) allows us to relate the two quantities in the proofs that follow.

---

[5]Notice that we consider the more general case of having $d$ number of rows (instead of $m$).

We now rephrase the above discussion in order to cast it in the framework of Randomized Linear Algebra *(RLA)* and active learning. Dropping the index $i$ for notational simplicity, we assume that $\mathbf{Az} \approx \mathbf{t}$ and we seek to recover an approximate solution vector $\mathcal{Q}(\mathbf{t})$ such that the $\ell_p$ norm error between the approximate and the optimal solution is bounded. *Importantly*, the query protocol $\mathcal{Q}$ is *not* allowed full access to the vector $\mathbf{t}$ in order to construct the approximate solution vector. This is a well-studied problem in the context of RLA: the learner is given a large collection of $k$-dimensional data points (the $d \gg k$ rows of the design matrix $\mathbf{A} \in \mathbb{R}^{d \times k}$), but can only query a small subset of the real-valued labels associated with each data point (elements of the vector $\mathbf{t} \in \mathbb{R}^d$). Prior work in RLA and active learning has studied this problem in order to identify the optimal number of queries that allow efficient, typically relative error, approximations of the loss function. In our parlance, prior work has explored the existence of query protocols that construct a vector $\mathcal{Q}(\mathbf{t})$ such that

$$\|\mathbf{t} - \mathbf{A}\mathcal{Q}(\mathbf{t})\|_p \leq \gamma_p \|\mathbf{t} - \mathbf{Az}\|_p, \tag{1}$$

where $\gamma_p > 1$ is an error parameter that controls the approximation accuracy. Of particular interest in the RLA literature are *relative error* approximations, with $\gamma_p = 1 + \epsilon$, for some small $\epsilon > 0$; see [MMWY21, MMWY22] for a detailed discussion. However, relative error approximations are less important in our setting, since our protocols in Section 3.2 necessitate $\zeta_p \geq \varepsilon_{\mathtt{mdl},p}$. For $p = 2$, the underlying problem is active learning for least-squares regression: [DMMS11] analyzed its complexity (namely, the number of queries) of query protocols in this setting, eventually providing matching upper and lower bounds. Similarly, for $p = 1$, the underlying problem is active learning for least absolute deviation regression, a robust version of least-squares regression: [MD21] analyzed the complexity of query protocols in this setting. The query protocols of [DMMS11, MD21] are straightforward: they sample a small set of labels (i.e., bidder types) and elicit the bidder's preferences for this set. Then, the respective $\ell_p$ norm regression problem is solved on the smaller set and the resulting solution is returned as an approximation to the original solution.[6] The types to be sampled (see Appendix D.1 for details) are selected using distributions that can be constructed by accessing *only* the design matrix $\mathbf{A}$. Specifically, for the $p = 2$ case, one needs to compute or approximate the *leverage scores* of the rows of the matrix $\mathbf{A}$. For the $p = 1$ case, one needs to compute or approximate the *Lewis weights* of the design matrix $\mathbf{A}$. (The Lewis weights are an extension of the leverage scores to $\ell_p$ norms for $p \neq 2$.) The work of [DW17, DWH18, DM21, DW18, CP19] for the $p = 2$ case involves more elaborate query protocols, using primitives such as volume sampling and the Batson-Spielman-Srivastava sparsifier to improve the query complexity. Finally, the $p > 2$ case for active learning for regression problems was recently resolved in [MMWY22, MMWY21]; we discuss their approach in our context in Appendix D.2.

To the best of our knowledge, our work is the first one to formulate connections between mechanism design and Randomized Linear Algebra for active learning. Two technical points of departure that are needed in order to adapt the RLA work for active learning to the mechanism design framework are: *(i)* we need to derive bounds of the form of eqn. (1) for the $\ell_p$ norm distance between the exact and approximate solutions, whereas prior work typically bounds the error of the *loss* function when an approximate solution is used; and *(ii)* the entries of the bidder's type vector $\mathbf{t}$ might not be known exactly, but only up to a small error. The latter assumption corresponds to the use of *noisy queries* in the model of [CD22] and is known to be equivalent, up to logarithmic factors, to *threshold queries* via binary search. Our work addresses both technicalities and seamlessly combines the RLA work for active learning with mechanism design.

Prior to stating our main result, we need to define a fundamental property of the design matrix $\mathbf{A} \in \mathbb{R}^{d \times k}$ that will affect the approximation error. Let

$$\sigma_{\min,p}(\mathbf{A}) = \min_{\mathbf{x} \in \mathbb{R}^k, \, \|\mathbf{x}\|_p = 1} \|\mathbf{Ax}\|_p. \tag{2}$$

For $p = 2$, this is simply the smallest singular value of the matrix $\mathbf{A}$. For other values of $p$, the above definition is the standard generalization of the smallest singular value of $\mathbf{A}$ for the induced matrix $\ell_p$ norm. Notice that $\sigma_{\min,p}(\mathbf{A})$ is a property of the matrix $\mathbf{A}$ and can be computed *a priori* via, say, the QR factorization or the Singular Value Decomposition (SVD) for $p = 2$ and via linear programming for $p = 1$. As we will see in Theorem 1 below, smaller values of $\sigma_{\min,p}(\mathbf{A})$ result in increased sample complexity for our query protocols.

---

[6]To be precise, multiple smaller problems have to be solved and a "good enough" solution has to be chosen in order to boost the success probability. See Appendix D for details.

**Theorem 1.** *Let $\mathbf{A} \in \mathbb{R}^{d \times k}$ be the design matrix, and recall the definitions of the model error $\varepsilon_{mdl,p}$ (Definition 5) and the query noise (Definition 4). For all integers $1 \le p < \infty$, there exist query protocols $\mathcal{Q}$ using $s_p$ queries for each bidder $i \in [n]$, such that, with probability at least $1 - \delta$,*

$$\|\mathbf{z}_i - \mathcal{Q}(\mathbf{t}_i)\|_p \le \frac{c_p(\varepsilon_{mdl,p} + \varepsilon_{nq,p})}{\sigma_{\min,p}(\mathbf{A})} = \zeta_p$$

*holds for all $n$ bidders $i \in [n]$. Here $c_p$ is a small constant that depends on $p$.[7] The respective query complexities for $p = 1$ and $p = 2$ are (asymptotically) identical:*

$$s_1 = s_2 = O\left(k \cdot \ln k \cdot \ln n/\delta\right). \tag{3}$$

*For $p \ge 3$, the query complexity is*

$$s_p = O\left(k^{p/2} \cdot \ln^3 k \cdot \ln n/\delta\right). \tag{4}$$

Several comments are in order. *(i)* The error $\zeta_p$ is a small constant times the modelling error plus the error due to noisy queries. In the limit case where the modelling error is equal to zero and the queries are noiseless, the bidders' types can be recovered exactly in our framework. However, as the modelling error and the query noise increase, approximating user types becomes harder and less accurate. *(ii)* Importantly, the approximation accuracy of Theorem 1 grows linearly with the inverse of the smallest $\ell_p$ norm singular value of the design matrix $\mathbf{A}$. Our results indicate that the approximation accuracy of the query model $\mathcal{Q}$ depends on this simple property of the design matrix $\mathbf{A}$. For example, focusing on the $p = 2$ case, our theorem shows that as the archetypes (columns of the matrix $\mathbf{A}$) become more linearly dependent and the smallest singular value approaches zero, the error of our approximation worsens. This is quite reasonable: if archetypes are linearly dependent, then it is increasingly difficult to approximate the respective entries of the vector $\mathbf{z}$. *(iii)* The query complexities $s_1$ and $s_2$ are asymptotically identical, growing linearly with $k \ln k$, where $k$ is the number of archetypes. They both depend on the log of the number of bidders (due to a union bound) and on the log of $1/\delta$, where $\delta$ is the failure probability. The query complexity for $p \ge 3$ is larger and is dominated by the $k^{p/2}$ term. Importantly, the query complexity remains independent of $d$, the number of bidder types, which, in worst case, could be exponential to the number of underlying items. *(iv)* Improving the sampling complexities $s_1$ and $s_2$ has been a topic of intense interest in the RLA community and we defer the reader to [MMWY21, MMWY22], which has essentially provided matching upper and lower bounds for various values of $p$. We just note that for the well-studied $p = 2$ case, volume sampling approaches [DW17, DWH18, DM21, DW18] achieve essentially matching bounds, while the work of [CP19] removes (at least in expectation) the $\ln k$ factor from $s_2$, at the expense of significant additional protocol complexity. From a technical perspective, we note that $\zeta_p \ge \varepsilon_{mdl,p}$, as necessitated in Theorem 2 and that our query protocols are all one-round protocols.

Finally, notice that our theorem works for all $p \ge 1$, but not for $p = \infty$, which was the setting of [CD22]. In Appendix D.4, we present a (modest) improvement of the result of [CD22] and explain why it seems improbable that the $p = \infty$ case can be generalized to a much broader class of design matrices. This is a strong motivating factor to explore properties of mechanism design for the recommender system setting for other $\ell_p$ norms, as we do in this work.

## 3.2 The Mechanism Design component

The goal of the mechanism design component is to transform a mechanism $\hat{\mathcal{M}}$ for $\hat{\mathcal{D}}$ into a mechanism $\mathcal{M}$ for $\mathcal{D}_z$. We first define exactly the type of access to $\hat{\mathcal{D}}_z$ and $\hat{\mathcal{M}}$ our construction requires.

**Definition 6** (Access to $\hat{\mathcal{M}}$). *By "query access to $\hat{\mathcal{M}}$" we mean access to an oracle which, given a valuation profile $\mathbf{t}$, outputs the allocation and payments of $\hat{\mathcal{M}}$ on input $\mathbf{t}$.*

**Definition 7** (Access to $\hat{\mathcal{D}}_z$). *By "oracle access to $\hat{\mathcal{D}}$" we mean access to (1) a sampling algorithm $\mathcal{S}_i$ for each $i \in [n]$, where $\mathcal{S}_i(\mathbf{x}, \delta)$ draws a sample from the conditional distribution of $\hat{\mathcal{D}}_{z,i}$ on the $k$-dimensional cube $\times_{j \in [k]}[x_j, x_j + \delta_j)$, and (2) an oracle which, given as input a type $\mathbf{t}_i$ for bidder $i$, outputs the type in the support of $\hat{\mathcal{D}}_{z,i}$ that is closest to $\mathbf{t}_i$ in $\ell_p$ distance, i.e., outputs $argmin_{\mathbf{t}_i' \in supp(\hat{\mathcal{D}}_{z,i})}\|\mathbf{t}_i - \mathbf{t}_i'\|_p$.*

---

[7]We make no attempt to optimize constants and focus on simplicity of presentation. In our proofs, $c_1 = 2.5$; $c_2 = 7.5$; and for $p \ge 3$, $c_p = 18 \cdot (200)^{1/p} + 3$. Notice that the last constant converges to 21 as $p$ increases.

If the allocation is randomized, our approach works even if the query to the oracle returns a (correct) deterministic instantiation of the randomized allocation.[8]

In Definition 7, the first part of our oracle access (sampling from the conditional) is also necessary in [CD22]. The second part is new to our work, and replaces a strong requirement in [CD22]. In more detail, given a type $\mathbf{t}_i$, Cai and Daskalakis [CD22] (as well as Brustle et al. [BCD20]) need to know if $\mathbf{t}_i \in supp(\widetilde{\mathcal{D}}_{z,i})$, and, if not, need access to $argmax_{\mathbf{t}'_i \in supp(\widetilde{\mathcal{D}}_{z,i})} \mathbb{E}_{\mathbf{b}_{-i} \sim \widetilde{\mathcal{D}}_{z,-i}}[u_i(v_i, \hat{\mathcal{M}}(\mathbf{t}'_i, \mathbf{b}_{-i}))]$ where $\widetilde{\mathcal{D}}_{z,i}$ is a rounded-down version of $\hat{\mathcal{D}}_{z,i}$. However, for arbitrary distributions and mechanisms, this task might be computationally inefficient, or simply infeasible. In our work, we need access to $argmin_{\mathbf{t}'_i \in supp(\hat{\mathcal{D}}_{z,i})} \|\mathbf{t}_i - \mathbf{t}'_i\|_p$ in the "no" case.

Given these definitions, our main theorem for this component is stated as follows.

**Theorem 2.** *Let $\mathcal{D} = \times_{i=1}^n \mathcal{D}_i$ be the bidders' type distribution and $v_i : \mathbb{R}^d \times 2^{[m]} \to \mathbb{R}_+$ be a $\mathcal{L}$-Lipschitz valuation function for each bidder $i \in [n]$. Also, let $\mathbf{A} \in \mathbb{R}^{d \times k}$ be a design matrix and $\hat{\mathcal{D}}_z = \times_{i=1}^n \hat{\mathcal{D}}_{z,i}$, where $\hat{\mathcal{D}}_{z,i}$ is a distribution over $\mathbb{R}^k$ for each $i \in [n]$.*

*Suppose that we are given (1) query access to a mechanism $\hat{\mathcal{M}}$ that is IR and BIC w.r.t. $\hat{\mathcal{D}}_z$ and valuations $\{v_i^{\mathbf{A}}\}_{i \in [n]}$, (2) oracle access to $\hat{\mathcal{D}}_z$, and (3) any $(\varepsilon_{\mathtt{mdl},p}, \zeta_p, p)$-query protocol $\mathcal{Q}$ with $\zeta_p \geq \varepsilon_{\mathtt{mdl},p}$. Then, we can construct a mechanism $\mathcal{M}$ that is oblivious to $\mathcal{D}$ and $v(\cdot, \cdot)$, such that for all $\mathcal{D}$ that satisfy $\pi_p(\mathcal{D}_i, \mathbf{A} \circ \hat{\mathcal{D}}_{z,i}) \leq \varepsilon_{\mathtt{mdl},p}$ for all $i \in [n]$, the following hold: (1) $\mathcal{M}$ only interacts with every bidder using $\mathcal{Q}$ once, (2) $\mathcal{M}$ is IR and $(\eta, \mu)$-BIC w.r.t. $\mathcal{D}$, where $\mu = O(\sqrt{\zeta_p})$ and $\eta = O(n\|\mathbf{A}\|_\infty \mathcal{L} \sqrt{\zeta_p})$, and (3) the expected revenue of $\mathcal{M}$ is at least $Rev(\hat{\mathcal{M}}, \hat{\mathcal{D}}_z) - O(n\eta)$.*

Note that $\mathcal{M}$ is an indirect mechanism, so it is slightly imprecise to call it $(\eta, \mu)$-BIC. Formally, interacting with $\mathcal{Q}$ truthfully is an approximate Bayes-Nash equilibrium.

In order to prove Theorem 2, we use a key lemma, Lemma 2, which establishes the robustness guarantees of Theorem 2, but in the space of latent types. Intuitively, let $\mathbf{t}_i$ be the type of bidder $i$, and $\mathbf{z}_i$ be a random variable distributed according to $\hat{\mathcal{D}}_{z,i}$. We know that $\pi_p(\mathcal{D}_i, \mathbf{A} \circ \hat{\mathcal{D}}_i) \leq \varepsilon_{\mathtt{mdl},p} \leq \zeta_p$. Due to Lemma 1, there exists a coupling such that with probability greater than $1 - \zeta_p$, $\|\mathbf{t}_i - \mathbf{A}\mathbf{z}_i\|_p \leq \zeta_p$. Since the seller uses a $(\varepsilon_{\mathtt{mdl},p}, \zeta_p, p)$-query protocol, with probability at least $1 - \zeta_p$, $\|\mathcal{Q}(\mathbf{t}_i) - \mathbf{z}_i\|_p \leq \zeta_p$. Note that, this implies that $\mathbf{z}_i$ and $\mathcal{Q}(t_i)$ are distributed such that their Prokhorov distance is at most $\zeta_p$. At this step, Lemma 2 provides us a mechanism $\widetilde{\mathcal{M}}$, constructed from $\hat{\mathcal{M}}$, that we can execute on types $\mathcal{Q}(\mathbf{t}_1), \dots, \mathcal{Q}(\mathbf{t}_n)$, obtained by interacting with the bidders via the query protocol. With probability at least $1 - \zeta_p$, we have that $\|\mathbf{t}_i - \mathbf{A}\mathbf{z}_i\|_\infty \leq \|\mathbf{t}_i - \mathbf{A}\mathbf{z}_i\|_p \leq \varepsilon_{\mathtt{mdl},p}$ and thus, using the fact that the query protocol ensures $\|\mathcal{Q}(\mathbf{t}_i) - \mathbf{z}_i\|_p \leq \zeta_p$ as well, we have $\|\mathbf{t}_i - \mathbf{A}\mathcal{Q}(\mathbf{t}_i)\|_\infty \leq \varepsilon_{\mathtt{mdl},p} + k\|\mathbf{A}\|_\infty \zeta_p$. The guarantees of $\widetilde{\mathcal{M}}$ for the distribution over $\mathcal{Q}(\mathbf{t}_i)$s are therefore translated into guarantees (with a small error) of the overall mechanism for the $\mathcal{D}$.

The proof of Lemma 2 is quite involved and is the main focus of our analysis. Here, we sketch the key ideas behind the proof, and defer all formal arguments to Appendix C. The proof uses the following notion of a rounded distribution.

**Definition 8** (Rounded Distribution). *Let $\mathcal{F}$ be a distribution supported on $\mathbb{R}_{\geq 0}^k$. For any $\delta > 0$ and $\ell \in [0, \delta]^k$, we define function $r^{(\ell, \delta)} : \mathbb{R}_{\geq 0}^k \mapsto \mathbb{R}_{\geq 0}^k$ such that $r_i^{(\ell, \delta)}(\mathbf{x}) = max\left\{\left\lfloor \frac{x_i - \ell_i}{\delta} \right\rfloor \cdot \delta + \ell_i, 0\right\}$ for all $i \in [k]$. Let $\mathbf{x}$ be a random vector sampled from $\mathcal{F}$. We define $\lfloor \mathcal{F} \rfloor_{\ell, \delta}$ as the distribution for the random variable $r^{(\ell, \delta)}(\mathbf{x})$, and we call $\lfloor \mathcal{F} \rfloor_{\ell, \delta}$ the rounded distribution of $\mathcal{F}$.*

We follow an approach similar to Brustle et al. [BCD20]. The main idea is that arguing directly about mechanisms for distributions that are close in Prokhorov distance is difficult. On the flip side, arguing about mechanisms for distributions that are close in total variation distance is much easier, since the total variation is a more stringent (and hence more tamable) notion of distance. The key observation is that, if two distributions are close in Prokhorov distance then, in expectation over the random parameter $\ell$, their rounded-down versions are close in total variation distance.

---

[8]That is, if, for example, $\hat{\mathcal{M}}$ allocates item $j$ to bidder $i$ with probability $1/2$ and with the remaining probability item $j$ is not allocated, our construction does not need to know this distribution/fractional allocation and works even if nature samples and returns an integral allocation for item $j$.

Our overall construction is via three reductions. First, in Lemma 3, given a mechanism for $\hat{\mathcal{F}}_z$ we design a mechanism for the rounded-down version. Second, in Lemma 4, given a mechanism for the rounded-down $\hat{\mathcal{F}}_z$ we design a mechanism for $\lfloor \mathcal{F}_z \rfloor_{\ell,\delta}$, which maintains its guarantees if $\pi_p(\mathcal{F}_z, \hat{\mathcal{F}}_z)$ is small. Third, in Lemma 5, given a mechanism for $\lfloor \mathcal{F}_z \rfloor_{\ell,\delta}$ we design a mechanism for $\mathcal{F}_z$. Fig. 1 presents a detailed overview of the overall design architecture, and how the RLA and different mechanism design components interact.

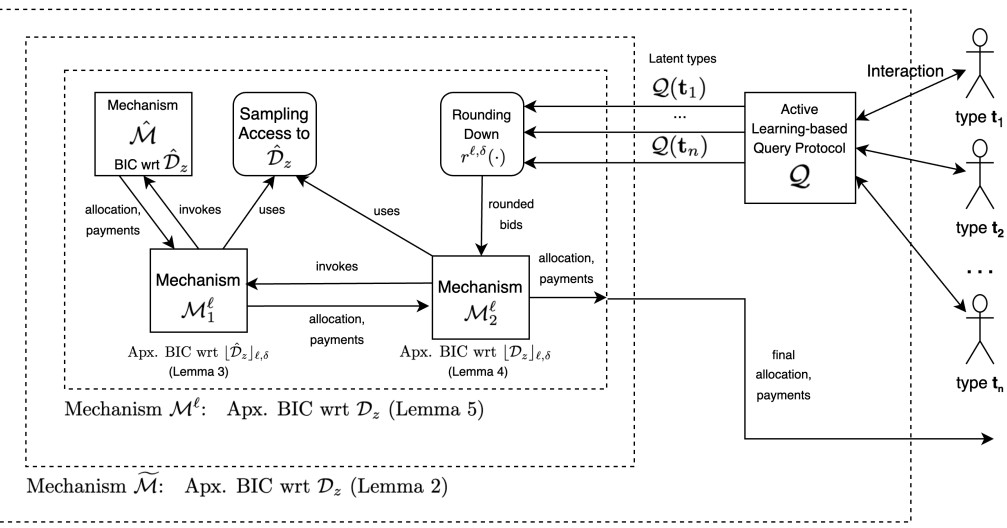

**Overall Design of the RLA for Active Learning-based Mechanism Design Framework**

Figure 1: Agents interact with the query protocol $\mathcal{Q}$, which learns their latent types $\mathcal{Q}(\mathbf{t}_i)$s. The mechanism design component (which is oblivious to the distributions of the true agents' types $\mathcal{D}$) then uses these to produce the final allocation and payments, utilizing only query access to $\hat{\mathcal{M}}$ and sampling access to $\hat{\mathcal{D}}_z$ such that the overall framework is approximately $(\eta, \mu)$-BIC wrt $\mathcal{D}$.

Our proofs for Lemmas 3 and 5 are adaptations of the corresponding lemmas of [BCD20], where our main task is to flesh out the exact dependence on the dimensionality of the latent space and the $\ell_p$-norm (versus the $\ell_1$-norm in [BCD20]). The novelty of our approach comes in the construction and analysis of Lemma 4. The difficulty of transforming mechanisms for $\lfloor \hat{\mathcal{F}}_z \rfloor_{\ell,\delta}$ into mechanisms for $\lfloor \mathcal{F}_z \rfloor_{\ell,\delta}$ is that the two distributions might not share the same support. Thus, we need a way to map bids that are not in the support of $\lfloor \hat{\mathcal{F}}_z \rfloor_{\ell,\delta}$ to bids that are. Brustle et al. [BCD20] do this by "optimally misreporting" on behalf of the bidder, by calculating $argmax_{\mathbf{z} \in supp(\lfloor \hat{\mathcal{F}}_{z,i} \rfloor_{\ell,\delta})} \mathbb{E}_{b_{-i} \sim \lfloor (\hat{\mathcal{F}}_z)_{-i} \rfloor_{\ell,\delta}} [u_i(v_i, \hat{\mathcal{M}}(\mathbf{z}, \mathbf{b}_{-i}))]$, and then picking matching payments that make the overall mechanism IR. Our approach leverages the fact that $\hat{\mathcal{F}}_{z,i}$ and $\mathcal{F}_{z,i}$ are close in Prokhorov distance, and thus any point on the support of one distribution is close to a point on the support of the other, with high probability. An ideal construction would map a report $\mathbf{w}_i$ to the "valid" report (i.e., a report in the support of $\lfloor \hat{\mathcal{F}}_{z,i} \rfloor_{\ell,\delta}$) that minimizes the $\ell_p$ distance to $\mathbf{w}_i$. This operation is linear on the support of $\lfloor \hat{\mathcal{F}}_{z,i} \rfloor_{\ell,\delta}$, and does not need any information on the valuation functions, nor on the actual probabilities that elements of the distribution are sampled with. Unfortunately, our assumption on what "oracle access" means does not allow us to do this operation (finding the closest point w.r.t. $\ell_p$) on $\lfloor \hat{\mathcal{F}}_{z,i} \rfloor_{\ell,\delta}$, but only on $\hat{\mathcal{F}}_{z,i}$; we prove that, by occurring a small loss, our assumption suffices.

### 3.3 Putting everything together

Combining Theorems 1 and 2, we can give mechanisms for concrete settings. Formally, we have the following theorem.

**Theorem 3.** *Under the same setting as in Theorem 2, for bidders with arbitrary valuation functions, we can construct mechanism $\mathcal{M}$ using only query access to the mechanism $\hat{\mathcal{M}}$ (Definition 6) and oracle access to distribution $\hat{\mathcal{D}}$ (Definition 7), and oblivious to the true type distribution $\mathcal{D}$. We consider queries (to each bidder $i$) of the form "What is your value for the subset of items $S$?"*

*Mechanism $\mathcal{M}$ is IR and $(\eta, \mu)$-BIC w.r.t. $\mathcal{D}$, where $\mu = O(\sqrt{\zeta_p})$ and $\eta = O(n\|\mathbf{A}\|_\infty \mathcal{L} \sqrt{\zeta_p})$, and the expected revenue of $\mathcal{M}$ is at least $Rev(\hat{\mathcal{M}}, \hat{\mathcal{D}}_z) - O(n\eta)$. Additionally, with probability at least $1 - \delta$,*

$$\zeta_p = c_p \cdot (\sigma_{\min,p}(\mathbf{A}))^{-1} \cdot \varepsilon_{\mathtt{mdl},p}$$

*for a small constant $c_p$ that depends on the parameter $p$ (see footnote 7). The number of queries is $O\left(k \cdot \ln k \cdot \ln(n/\delta)\right)$ and $O\left(k^{p/2} \cdot \ln^3 k \cdot \ln(n/\delta)\right)$ for $p = 1, 2$ and $p \geq 3$, respectively.*

The proof of Theorem 3 follows from Theorem 2 and Theorem 1, and is deferred to Appendix B.

As we've already discussed, the main mechanism of Cai and Daskalakis [CD22] requires bidders to have constrained-additive valuations[9], as well $\mathbf{A}$ to satisfy a number of restrictions. Here, we completely remove both conditions. On the flip-side, [CD22] ask bidders weaker queries, of the form "are you willing to pay price $\tau$ for item $j$?" Using such queries, one can binary search over $\tau$, and drive down the query noise (see Definition 4). For $\ell_\infty$, the extra cost of such an operation would be $\ln(\|\mathbf{A}\|_\infty/\varepsilon)$, where $\varepsilon$ is the desired accuracy. However, for other $p$-norms, for the same target accuracy, this operation requires an extra factor of $\Theta(\log(d^{1/p}))$ queries, giving a dependence on $d$.

## 4 Conclusions and Future Work

In this paper, we study mechanism design for prior distributions close to a topic model, inspired from the recommender systems literature. We formulate connections between mechanism design and Randomized Linear Algebra for active learning for regression problems, importing state-of-the-art results from Randomized Linear Algebra to mechanism design, and alleviate or relax restrictive assumptions of prior work. Developing a deeper understanding of such connections is an important direction for future research. For example, one could study this and other topic models in the context of mechanism design for correlated bidders, two-sided markets, information structure design, etc. Additionally, another interesting open problem would be to develop a framework for proving lower bounds for mechanism design (e.g., lower bounds on the query complexity for single-round or multi-round protocols used to communicate with the bidders) using known limitations of algorithms in active learning, and vice-versa.

**Acknowledgements**

Christos Boutsikas, Petros Drineas and Marios Mertzanidis are supported in part by a DOE award SC0022085, and NSF awards CCF-1814041, CCF-2209509, and DMS-2152687. Alexandros Psomas and Paritosh Verma are supported in part by an NSF CAREER award CCF-2144208, a Google Research Scholar Award, and a Google AI for Social Good award.

---

[9]A valuation function is constrained-additive if $v(t, S) = max_{T \in \mathcal{I} \cap 2^S} \sum_{j \in T} t_j$, where $\mathcal{I}$ is a downward-closed set system.

