# References

[AFH+12] Saeed Alaei, Hu Fu, Nima Haghpanah, Jason Hartline, and Azarakhsh Malekian. Bayesian optimal auctions via multi-to single-agent reduction. In *Proceedings of the 13th ACM Conference on Electronic Commerce*, pages 17–17, 2012.

[BCD20] Johannes Brustle, Yang Cai, and Constantinos Daskalakis. Multi-item mechanisms without item-independence: Learnability via robustness. In *Proceedings of the 21st ACM Conference on Economics and Computation*, EC '20, page 715–761, New York, NY, USA, 2020. Association for Computing Machinery.

[BCKW15] Patrick Briest, Shuchi Chawla, Robert Kleinberg, and S Matthew Weinberg. Pricing lotteries. *Journal of Economic Theory*, 156:144–174, 2015.

[BILW20] Moshe Babaioff, Nicole Immorlica, Brendan Lucier, and S Matthew Weinberg. A simple and approximately optimal mechanism for an additive buyer. *Journal of the ACM (JACM)*, 67(4):1–40, 2020.

[CD17] Yang Cai and Constantinos Daskalakis. Learning multi-item auctions with (or without) samples. In *2017 IEEE 58th Annual Symposium on Foundations of Computer Science (FOCS)*, pages 516–527. IEEE, 2017.

[CD22] Yang Cai and Constantinos Daskalakis. Recommender systems meet mechanism design. In *Proceedings of the 23rd ACM Conference on Economics and Computation*, EC '22, page 897–914, New York, NY, USA, 2022. Association for Computing Machinery.

[CDW12a] Yang Cai, Constantinos Daskalakis, and S Matthew Weinberg. An algorithmic characterization of multi-dimensional mechanisms. In *Proceedings of the forty-fourth annual ACM symposium on Theory of computing*, pages 459–478, 2012.

[CDW12b] Yang Cai, Constantinos Daskalakis, and S Matthew Weinberg. Optimal multi-dimensional mechanism design: Reducing revenue to welfare maximization. In *2012 IEEE 53rd Annual Symposium on Foundations of Computer Science*, pages 130–139. IEEE, 2012.

[CDW13a] Yang Cai, Constantinos Daskalakis, and S Matthew Weinberg. Reducing revenue to welfare maximization: Approximation algorithms and other generalizations. In *Proceedings of the Twenty-Fourth Annual ACM-SIAM Symposium on Discrete Algorithms*, pages 578–595. SIAM, 2013.

[CDW13b] Yang Cai, Constantinos Daskalakis, and S Matthew Weinberg. Understanding incentives: Mechanism design becomes algorithm design. In *2013 IEEE 54th Annual Symposium on Foundations of Computer Science*, pages 618–627. IEEE, 2013.

[CDW16] Yang Cai, Nikhil R Devanur, and S Matthew Weinberg. A duality based unified approach to bayesian mechanism design. In *Proceedings of the forty-eighth annual ACM symposium on Theory of Computing*, pages 926–939, 2016.

[CHK07] Shuchi Chawla, Jason D. Hartline, and Robert Kleinberg. Algorithmic pricing via virtual valuations. In *Proceedings of the 8th ACM Conference on Electronic Commerce*, EC '07, pages 243–251, New York, NY, USA, 2007. ACM.

[CHMS10] Shuchi Chawla, Jason D Hartline, David L Malec, and Balasubramanian Sivan. Multi-parameter mechanism design and sequential posted pricing. In *Proceedings of the forty-second ACM symposium on Theory of computing*, pages 311–320. ACM, 2010.

[CM16] Shuchi Chawla and J Benjamin Miller. Mechanism design for subadditive agents via an ex ante relaxation. In *Proceedings of the 2016 ACM Conference on Economics and Computation*, pages 579–596. ACM, 2016.

[CMS15] Shuchi Chawla, David Malec, and Balasubramanian Sivan. The power of randomness in bayesian optimal mechanism design. *Games and Economic Behavior*, 91:297–317, 2015.

[CP15]  Michael B Cohen and Richard Peng. $\ell_p$ Row Sampling by Lewis Weights. In *Proceedings of the forty-seventh annual ACM symposium on Theory of computing*, pages 183–192, 2015.

[CP19]  Xue Chen and Eric Price. Active regression via linear-sample sparsification. In *Conference on Learning Theory*, pages 663–695. PMLR, 2019.

[CR14]  Richard Cole and Tim Roughgarden. The sample complexity of revenue maximization. In *Proceedings of the forty-sixth annual ACM symposium on Theory of computing*, pages 243–252, 2014.

[CZ17]  Yang Cai and Mingfei Zhao. Simple mechanisms for subadditive buyers via duality. In *Proceedings of the 49th Annual ACM SIGACT Symposium on Theory of Computing*, pages 170–183. ACM, 2017.

[Das15]  Constantinos Daskalakis. Multi-item auctions defying intuition? *ACM SIGecom Exchanges*, 14(1):41–75, 2015.

[DDT13]  Constantinos Daskalakis, Alan Deckelbaum, and Christos Tzamos. Mechanism design via optimal transport. In *Proceedings of the fourteenth ACM conference on Electronic commerce*, pages 269–286, 2013.

[DDT15]  Constantinos Daskalakis, Alan Deckelbaum, and Christos Tzamos. Strong duality for a multiple-good monopolist. In *Proceedings of the Sixteenth ACM Conference on Economics and Computation*, pages 449–450, 2015.

[DHP16]  Nikhil R Devanur, Zhiyi Huang, and Christos-Alexandros Psomas. The sample complexity of auctions with side information. In *Proceedings of the forty-eighth annual ACM symposium on Theory of Computing*, pages 426–439, 2016.

[DM16]  Petros Drineas and Michael W Mahoney. RandNLA: Randomized Numerical Linear Algebra. *Communications of the ACM*, 59(6):80–90, 2016.

[DM21]  Michał Derezinski and Michael W Mahoney. Determinantal point processes in randomized numerical linear algebra. *Notices of the American Mathematical Society*, 68(1):34–45, 2021.

[DMMS11]  Petros Drineas, Michael W Mahoney, Shan Muthukrishnan, and Tamás Sarlós. Faster least squares approximation. *Numerische Mathematik*, 117(2):219–249, 2011.

[DS03]  David Donoho and Victoria Stodden. When does non-negative matrix factorization give a correct decomposition into parts? *Advances in neural information processing systems*, 16, 2003.

[DW17]  Michal Derezinski and Manfred KK Warmuth. Unbiased estimates for linear regression via volume sampling. *Advances in Neural Information Processing Systems*, 30, 2017.

[DW18]  Michał Dereziński and Manfred K. Warmuth. Reverse iterative volume sampling for linear regression. *Journal of Machine Learning Research*, 19(23):1–39, 2018.

[DWH18]  Michal Derezinski, Manfred K Warmuth, and Daniel Hsu. Tail bounds for volume sampled linear regression. *arXiv preprint arXiv:1802.06749*, 2018.

[GHZ19]  Chenghao Guo, Zhiyi Huang, and Xinzhi Zhang. Settling the sample complexity of single-parameter revenue maximization. In *Proceedings of the 51st Annual ACM SIGACT Symposium on Theory of Computing*, pages 662–673, 2019.

[GW21]  Yannai A Gonczarowski and S Matthew Weinberg. The sample complexity of up-to-$\varepsilon$ multi-dimensional revenue maximization. *Journal of the ACM (JACM)*, 68(3):1–28, 2021.

[HMR15]  Zhiyi Huang, Yishay Mansour, and Tim Roughgarden. Making the most of your samples. In *Proceedings of the Sixteenth ACM Conference on Economics and Computation*, pages 45–60, 2015.

[HN19] Sergiu Hart and Noam Nisan. Selling multiple correlated goods: Revenue maximization and menu-size complexity. *Journal of Economic Theory*, 183:991–1029, 2019.

[HR15] Sergiu Hart and Philip J Reny. Maximal revenue with multiple goods: Nonmonotonicity and other observations. *Theoretical Economics*, 10(3):893–922, 2015.

[KW19] Robert Kleinberg and S Matthew Weinberg. Matroid prophet inequalities and applications to multi-dimensional mechanism design. *Games and Economic Behavior*, 113:97–115, 2019.

[LPW09] David A. Levin, Yuval Peres, and Elizabeth L. Wilmer. *Markov Chains and Mixing Times*. American Mathematical Society, Providence, RI, USA, first edition, 2009.

[M. 11] M. W. Mahoney. *Randomized algorithms for matrices and data*. Foundations and Trends in Machine Learning. NOW Publishers, Boston, 2011.

[MD21] Xue Musco and Michal Derezinski. Query complexity of least absolute deviation regression via robust uniform convergence. In *Conference on Learning Theory*, pages 1144–1179. PMLR, 2021.

[MMWY21] Cameron Musco, Christopher Musco, David P Woodruff, and Taisuke Yasuda. Active linear regression for $\ell_p$ norms and beyond. *arXiv preprint arXiv:2111.04888*, 2021.

[MMWY22] Cameron Musco, Christopher Musco, David P Woodruff, and Taisuke Yasuda. Active linear regression for $\ell_p$ norms and beyond. In *2022 IEEE 63rd Annual Symposium on Foundations of Computer Science (FOCS)*, pages 744–753. IEEE, 2022.

[MR16] Jamie Morgenstern and Tim Roughgarden. Learning simple auctions. In *Conference on Learning Theory*, pages 1298–1318. PMLR, 2016.

[MV07] Alejandro M Manelli and Daniel R Vincent. Multidimensional mechanism design: Revenue maximization and the multiple-good monopoly. *Journal of Economic theory*, 137(1):153–185, 2007.

[Mye81] Roger B Myerson. Optimal auction design. *Mathematics of operations research*, 6(1):58–73, 1981.

[PSCW22] Alexandros Psomas, Ariel Schvartzman Cohenca, and S Weinberg. On infinite separations between simple and optimal mechanisms. *Advances in Neural Information Processing Systems*, 35:4818–4829, 2022.

[RW15] Aviad Rubinstein and S Matthew Weinberg. Simple mechanisms for a subadditive buyer and applications to revenue monotonicity. In *Proceedings of the Sixteenth ACM Conference on Economics and Computation*, pages 377–394. ACM, 2015.

[Str65] V. Strassen. The Existence of Probability Measures with Given Marginals. *The Annals of Mathematical Statistics*, 36(2):423 – 439, 1965.

[Woo14] David P. Woodruff. Sketching as a Tool for Numerical Linear Algebra. *Foundations and Trends in Theoretical Computer Science*, 10(1-2):1–157, 2014.

[Yao15] Andrew Chi-Chih Yao. An n-to-1 bidder reduction for multi-item auctions and its applications. In *Proceedings of the Twenty-Sixth Annual ACM-SIAM Symposium on Discrete Algorithms*, pages 92–109. Society for Industrial and Applied Mathematics, 2015.

# A  Notation

Table 1: Notations

| Symbol | Explanation |
|---|---|
| $n$ | Number of bidders |
| $m$ | Number of items |
| $k$ | Dimensionality of latent space |
| $d$ | Number of types |
| $\mathcal{D}_i$ | Distribution of bidders $i$ type vector |
| $\hat{\mathcal{D}}_i$ | Induced noisy representation of $D_i$ |
| $\hat{\mathcal{D}}_{z,i}$ | Distribution of latent vector type of bidder $i$ |
| $\mathbf{t}_i \in \mathbb{R}^d$ | Type vector of bidder $i$ |
| $\mathbf{z}_i \in \mathbb{R}^k$ | Sample from $\hat{\mathcal{D}}_{z,i}$ of bidder $i$ |
| $\mathbf{A} \in \mathbb{R}^{d \times k}$ | Design matrix of archetypes |
| $\mathcal{Q}(\mathbf{t}_i) = \tilde{\mathbf{z}}_i \in \mathbb{R}^k$ | Output of query protocol under true type $\mathbf{t}_i$ |
| $\boldsymbol{\epsilon}_{\mathtt{mdl},\mathbf{p}} \in \mathbb{R}^d$ | Modeling vector noise |
| $\varepsilon_{\mathtt{mdl},p} > 0$ | Upper bound on $\|\boldsymbol{\epsilon}_{\mathtt{mdl},\mathbf{p}}\|_p$ |
| $\boldsymbol{\epsilon}_{\mathtt{nq},\mathbf{p}} \in \mathbb{R}^d$ | Query vector noise |
| $\varepsilon_{\mathtt{nq},p} \geq 0$ | Upper bound on $\|\boldsymbol{\epsilon}_{\mathtt{nq},\mathbf{p}}\|_p$ |
| $v_i(\cdot,\cdot)$ | Valuation function |
| $\mathcal{M}$ | Mechanism |
| $x(\cdot)$ | Allocation rule |
| $p(\cdot)$ | Payment rule |
| $Rev(\mathcal{M},\mathcal{D})$ | Expected revenue of mechanism $M$ under type distributions $D$ |
| $\pi_P(\cdot,\cdot)$ | Prokhorov distance of two probability measures |
| $\gamma$ | A coupling between two probability measures |
| $v_i^{\mathbf{A}}(\mathbf{z}_i, S)$ | $v_i(\mathbf{A}\mathbf{z}_i, S)$ |
| $\mathcal{L}$ | Lipschitz constant of valuation function |
| $\lfloor F \rfloor_{\lambda,\delta}$ | Rounded down distribution |
| $r^{(\lambda,\delta)}(x)$ | Rounding down mapping function where $\delta$ is a parameter chosen by the designer |
| $\mathrm{supp}(F)$ | Support of probability distribution $F$ |
| $\rho^\ell$ | Total TV distance of rounded down distributions |
| $u_i(\mathbf{t}_i \leftarrow \mathbf{b}_i, \mathbf{t}_{-i})$ | Change in utility of bidder $i$ when bidder reports $\mathbf{b}_i$ instead of $\mathbf{t}_i$ and the remaining bidders bid $\mathbf{t}_{-i}$ |

# B  Proof of Theorem 3

*Proof of Theorem 3.* For arbitrary valuation functions, $d = 2^m$, and each type $\mathbf{t}_i$ represents the valuation of bidder $i$ for all possible bundles of items. Since we can ask queries of the form "What is your value for subset S?", in Theorem 1, we have that $\varepsilon_{\mathtt{nq},\mathbf{p}} = 0$. Thus from Theorem 1 with probability at least $1 - \delta$ we have a $\left(\varepsilon_{\mathtt{mdl},p}, \frac{c_p \varepsilon_{\mathtt{mdl},p}}{\sigma_{\min,p}(\mathbf{A})}, p\right)$-query protocol that communicates once with the bidders and asks a total of $s_p$ queries. Where $s_1 = s_2 = O\left(k \cdot \ln k \cdot \ln n/\delta\right)$ and $s_p = O\left(k^{p/2} \cdot \ln^3 k \cdot \ln n/\delta\right)$. Combining this observation with Theorem 2 proves the theorem. $\square$

# C  The Mechanism Design Component

The main goal of this section is to prove Theorem 2. In order to establish the theorem, we use the following key lemma, which essentially establishes the robustness guarantees described in Theorem 2, but in the space of latent types.

**Lemma 2.** *Let $\mathbf{A} \in \mathbb{R}^{d \times k}$ be a design matrix. Suppose there exists a collection of distributions over latent types $\{\hat{\mathcal{F}}_{z,i}\}_{i \in [n]}$, where the support of each $\hat{\mathcal{F}}_{z,i}$ lies in $[0,1]^k$, and a mechanism $\hat{\mathcal{M}}$ that is IR and BIC w.r.t. $\hat{\mathcal{F}}_z = \times_{i=1}^n \hat{\mathcal{F}}_{z,i}$ and valuations $\{v_i^{\mathbf{A}}\}_{i \in [n]}$, where each $v_i$ is an $\mathcal{L}$-Lipschitz valuation. Let $\mathcal{F}_z = \times_{i=1}^n \mathcal{F}_{z,i}$ be any distribution such that $\pi_p(\mathcal{F}_{z,i}, \hat{\mathcal{F}}_{z,i}) \leq \zeta_p$ for all $i \in [n]$. Given query access to $\hat{\mathcal{M}}$ (as defined in Definition 6), and sampling access to $\hat{\mathcal{F}}_z$ (as defined in Definition 7), we can construct a mechanism $\widetilde{\mathcal{M}}$ such that the following hold: (i) $\widetilde{\mathcal{M}}$ is IR and $(\eta, \mu)$-BIC w.r.t. $\mathcal{F}_z$, and (ii) the expected revenue of $\widetilde{\mathcal{M}}$ is $Rev(\widetilde{\mathcal{M}}, \mathcal{F}_z) \geq Rev(\hat{\mathcal{M}}, \hat{\mathcal{F}}_z) - O(n\eta)$, where $\mu = O(\zeta_p + \delta\, k^{1/p})$ and $\eta = O\left(k\mathcal{L}\|\mathbf{A}\|_\infty \left(\delta + n\left(1 + \frac{k^{1-1/p}}{\delta}\right)\zeta_p\right)\right)$, for all $\delta > 0$.*

The proof of Lemma 2 is quite involved and will be the main focus of the subsequent analysis. Before proving Lemma 2, we will show that it is sufficient to prove our main result, Theorem 2.

*Proof of Theorem 2.* Let $\mathbf{t}_i$ be the type of bidder $i$, and $\mathbf{z}_i$ be a random variable distributed according to $\hat{\mathcal{D}}_{z,i}$. We know that $\pi_p(\mathcal{D}_i, \mathbf{A} \circ \hat{\mathcal{D}}_i) \leq \varepsilon_{\mathtt{mdl},p} \leq \zeta_p$. Due to Lemma 1, there exists a coupling such that with probability greater than $1 - \zeta_p$, $\|\mathbf{t}_i - \mathbf{A}\mathbf{z}_i\|_p \leq \zeta_p$. Since the seller uses a $(\varepsilon_{\mathtt{mdl},p}, \zeta_p, p)$-query protocol, with probability at least $1 - \zeta_p$, $\|\mathcal{Q}(\mathbf{t}_i) - \mathbf{z}_i\|_p \leq \zeta_p$. Note that, this implies that $\mathbf{z}_i$ and $\mathcal{Q}(\mathbf{t}_i)$ are distributed such that their Prokhorov distance is at most $\zeta_p$.

We now invoke Lemma 2 by setting $\mathcal{F}_{z,i}$ to be distributed as $\mathcal{Q}(\mathbf{t}_i)$, $\hat{\mathcal{F}}_{z,i}$ to be distributed as $\hat{\mathcal{D}}_{z,i}$, and $\delta = \sqrt{\zeta_p}$. We run the resultant mechanism, $\widetilde{\mathcal{M}}$, on types $\mathcal{Q}(\mathbf{t}_1), \ldots, \mathcal{Q}(\mathbf{t}_n)$, obtained by interacting with the bidders via the query protocol. With probability at least $1 - \zeta_p$, we have that $\|\mathbf{t}_i - \mathbf{A}\mathbf{z}_i\|_\infty \leq \|\mathbf{t}_i - \mathbf{A}\mathbf{z}_i\|_p \leq \varepsilon_{\mathtt{mdl},p}$ and thus, using the fact that the query protocol ensures $\|\mathcal{Q}(\mathbf{t}_i) - \mathbf{z}_i\|_p \leq \zeta_p$ as well, we have $\|\mathbf{t}_i - \mathbf{A}\mathcal{Q}(\mathbf{t}_i)\|_\infty \leq \varepsilon_{\mathtt{mdl},p} + k\|\mathbf{A}\|_\infty \zeta_p$. Note that $\widetilde{\mathcal{M}}$ is a $(\eta, \mu)$-BIC mechanism w.r.t. $\times_{i \in [n]} \mathcal{F}_{z,i}$, with $\eta = O(n\|\mathbf{A}\|_\infty \mathcal{L}\sqrt{\zeta_p})$ and $\mu = \sqrt{\zeta_p}$. Therefore, conditioned on the aforementioned scenario having probability at least $1 - \zeta_p$, with probability at least $1 - \mu$, deviating from interacting with $\mathcal{Q}$ truthfully can increase the expected utility by at most $O(\mathcal{L}(\varepsilon_{\mathtt{mdl},p} + k\|\mathbf{A}\|_\infty \zeta_p) + \eta) = O(\eta)$, and bidders can improve their utility by deviating with probability less than $\varepsilon_{\mathtt{mdl},p} \leq \zeta_p$, i.e., with probability at most $1 - (1 - \zeta_p)(1 - \mu) = O(\zeta_p + \mu) = O(\mu)$. Hence, the overall mechanism is $(\eta, \mu)$-BIC w.r.t. $\mathcal{D}$, as well as IR (since $\widetilde{\mathcal{M}}$ is IR). The revenue guarantee is immediate. This concludes the proof. $\qquad\square$

### C.1 Proof of Lemma 2

For the remainder of this section, we use the following notation, where all distributions of an agent $i$ are supported on $[0,1]^k$: $\hat{\mathcal{F}}_z = \times_{i \in [n]} \hat{\mathcal{F}}_{z,i}$, $\left\lfloor \hat{\mathcal{F}}_z \right\rfloor_{\ell,\delta} = \times_{i \in [n]} \left\lfloor \hat{\mathcal{F}}_{z,i} \right\rfloor_{\ell,\delta}$, $\lfloor \mathcal{F}_z \rfloor_{\ell,\delta} = \times_{i \in [n]} \lfloor \mathcal{F}_{z,i} \rfloor_{\ell,\delta}$, and $\mathcal{F}_z = \times_{i \in [n]} \mathcal{F}_{z,i}$. We follow an approach similar to Brustle et al. [BCD20]. The main idea is that arguing directly about mechanisms for distributions that are close in Prokhorov distance is difficult. On the flip side, arguing about mechanisms for distributions that are close in total variation distance is much easier, since the total variation is a more stringent — and hence more well-behaved — notion of distance. The key observation here is that if two distributions are close in Prokhorov distance then, in expectation, their rounded-down versions will also be close in total variation distance, where the expectation is taken over the randomness of parameter $\ell$.

Our overall construction of the mechanism $\widetilde{\mathcal{M}}$ (of Lemma 2) is via three reductions. For ease of presentation, we defer some proofs (and spefically, the proofs for Lemmas 3, 4 and 5, to Appendix C.2. First, in Lemma 3, given a mechanism for $\hat{\mathcal{F}}_z$ we design a mechanism for the rounded-down version.

**Lemma 3.** *Given query access to a mechanism $\mathcal{M}$ that is IR and BIC w.r.t. $\hat{\mathcal{F}}_z$, and a sampling algorithm $\mathcal{S}_i$ for each $i \in [n]$, where $\mathcal{S}_i(\mathbf{x}, \delta)$ draws a sample from the conditional distribution of $\hat{\mathcal{F}}_{z,i}$ on the $k$-dimensional cube $\times_{j \in [k]}[x_j, x_j + \delta_j)$, we can construct a mechanism $\mathcal{M}_1^\ell$, oblivious to $\hat{\mathcal{F}}$, that is IR and $O(k\|\mathbf{A}\|_\infty \mathcal{L}\delta)$-BIC w.r.t. $\left\lfloor \hat{\mathcal{F}}_z \right\rfloor_{\ell,\delta}$. Furthermore,*

$$Rev\left(\mathcal{M}_1^\ell, \left\lfloor \hat{\mathcal{F}}_z \right\rfloor_{\ell,\delta}\right) \geq Rev(\mathcal{M}, \hat{\mathcal{F}}_z) - O(kn\|\mathbf{A}\|_\infty \mathcal{L}\delta).$$

Second, in Lemma 4, given a mechanism for $\left\lfloor \hat{\mathcal{F}}_z \right\rceil_{\ell,\delta}$ we design a mechanism for $\lfloor \mathcal{F}_z \rfloor_{\ell,\delta}$, which maintains its guarantees if $\pi_p(\mathcal{F}_z, \hat{\mathcal{F}}_z)$ is small.

**Lemma 4.** *Let $\left\lfloor \hat{\mathcal{F}}_z \right\rceil_{\ell,\delta}$ and $\lfloor \mathcal{F}_z \rfloor_{\ell,\delta}$ be distributions such that, for all $i \in [n]$, it holds that (1) $\| \left\lfloor \hat{\mathcal{F}}_{z,i} \right\rceil_{\ell,\delta} - \lfloor \mathcal{F}_{z,i} \rfloor_{\ell,\delta} \|_{TV} \leq \varepsilon_i^\ell$, and (2) $\pi_p\left(\hat{\mathcal{F}}_{z,i}, \mathcal{F}_{z,i}\right) \leq \zeta_p$. Then, letting $\rho^\ell := \sum_i \varepsilon_i^\ell$, given a mechanism $\mathcal{M}_1^\ell$ that is IR and $\eta$-BIC w.r.t $\left\lfloor \hat{\mathcal{F}}_z \right\rceil_{\ell,\delta}$ we can construct a mechanism $\mathcal{M}_2^\ell$ that is IR, $\left(O(k\mathcal{L}\|\mathbf{A}\|_\infty \rho^\ell + k\left(\zeta_p + \delta \cdot k^{1/p}\right) \|\mathbf{A}\|_\infty \mathcal{L} + \eta), \zeta_p + \delta k^{1/p}\right)$-BIC w.r.t. $\lfloor \mathcal{F}_z \rfloor_{\ell,\delta}$. Furthermore:*

$$Rev\left(\mathcal{M}_2^\ell, \lfloor \mathcal{F}_z \rfloor_{\ell,\delta}\right) \geq Rev\left(\mathcal{M}_1^\ell, \left\lfloor \hat{\mathcal{F}}_z \right\rceil_{\ell,\delta}\right) - nk\mathcal{L}\|\mathbf{A}\|_\infty \rho^\ell - nk(\zeta_p + \delta k^{1/p})\|\mathbf{A}\|_\infty \mathcal{L}.$$

Third, in Lemma 5, given a mechanism for $\lfloor \mathcal{F}_z \rfloor_{\ell,\delta}$ we design a mechanism for $\mathcal{F}_z$.

**Lemma 5.** *Given a mechanism $\mathcal{M}_2^\ell$ that is IR and $(\eta, \mu)$-BIC w.r.t. $\lfloor \mathcal{F}_z \rfloor_{\ell,\delta}$, we can construct a mechanism $\mathcal{M}^\ell$ that is IR and $(3k\mathcal{L}\|\mathbf{A}\|_\infty \delta + \eta, \mu)$-BIC w.r.t. $\mathcal{F}_z$. Moreover, $Rev\left(\mathcal{M}^\ell, \mathcal{F}_z\right) \geq Rev\left(\mathcal{M}_2^\ell, \lfloor \mathcal{F}_z \rfloor_{\ell,\delta}\right) - nk\mathcal{L}\|\mathbf{A}\|_\infty \delta$.*

With all the prerequisite technical lemmas at hand, we finally prove Lemma 2.

*Proof of Lemma 2.* The mechanism $\widetilde{\mathcal{M}}$ simply operates as follows: $(i)$ $\ell$ is sampled uniformly from the interval $[0, \delta]$ (i.e., $\ell \sim \mathcal{U}[0, \delta]$) and $(ii)$ the mechanism $\mathcal{M}^\ell$ of Lemma 5 is run on the input bids. Specifically, $\widetilde{\mathcal{M}}$ calls mechanism $\mathcal{M}^\ell$ of Lemma 5, which calls mechanism $\mathcal{M}_2^\ell$ of Lemma 4, which calls mechanism $\mathcal{M}_1^\ell$ of Lemma 3, which calls $\hat{\mathcal{M}}$ which is IR and BIC w.r.t. $\hat{\mathcal{F}}$. In order to invoke Lemma 4 we need a bound on the TV distance between $\left\lfloor \hat{\mathcal{F}}_{z,i} \right\rceil_{\ell,\delta}$ and $\lfloor \mathcal{F}_{z,i} \rfloor_{\ell,\delta}$. We use the following lemma (whose proof can be found in Appendix C.2).

**Lemma 6.** *Let $\mathcal{F}_z$ and $\hat{\mathcal{F}}_z$ be two distributions supported on $\mathbb{R}^k$ such that $\pi_p(\mathcal{F}_z, \hat{\mathcal{F}}_z) \leq \varepsilon$. For any $\delta > 0$, $\mathbb{E}_{\ell \sim \mathcal{U}[0,\delta]^k}\left[\| \lfloor \mathcal{F}_z \rfloor_{\ell,\delta} - \left\lfloor \hat{\mathcal{F}}_z \right\rceil_{\ell,\delta} \|_{TV}\right] \leq \left(1 + \frac{k^{1-1/p}}{\delta}\right)\varepsilon$.*

Letting $\rho^\ell = \sum_i \varepsilon_i^\ell$, where $\| \left\lfloor \hat{\mathcal{F}}_{z,i} \right\rceil_{\ell,\delta} - \lfloor \mathcal{F}_{z,i} \rfloor_{\ell,\delta} \|_{TV} \leq \varepsilon_i^\ell$, Lemma 6 implies that $\mathbb{E}_{\ell \sim \mathcal{U}[0,\delta]}\left[\rho^\ell\right] \leq n\left(1 + \frac{k^{1-1/p}}{\delta}\right)\zeta_p$, allowing us to invoke Lemma 4. The IR guarantee for $\widetilde{\mathcal{M}}$ is immediate. For the BIC guarantee, there is a trade-off in choosing $\delta$: $\mathbb{E}_{\ell \sim \mathcal{U}[0,\delta]}\left[\rho^\ell\right]$ has a term inversely proportional to $\delta$, while the BIC guarantees of Lemma 4 and Lemma 3 have a term proportional to $\delta$. We set $\delta = \sqrt{\zeta_p}$ to strike a balance between these terms. Combining the BIC and revenue guarantees of Lemma 3, Lemma 4, and Lemma 5, and using the fact that $k$ is a constant, we get that $\widetilde{\mathcal{M}}$ is a $(\eta, \mu)$-BIC and IR w.r.t. $\mathcal{F}$ where $\mu = O(\zeta_p + \delta \cdot k^{1/p})$ and $\eta = O\left(k\mathcal{L}\|\mathbf{A}\|_\infty \left(\delta + n \cdot \left(1 + \frac{k^{1-1/p}}{\delta}\right) \cdot \zeta_p\right)\right)$ and $Rev(\widetilde{\mathcal{M}}, \mathcal{F}) \geq Rev(\hat{\mathcal{M}}, \hat{\mathcal{F}}) - O(n\eta)$. $\qquad\square$

### C.2 Proofs of Lemma 3, 4, 5, and 6

The following proposition will be useful in multiple proofs throughout this section.

**Proposition 1.** *If $v_i$ is a $\mathcal{L}$-Lipschitz valuation function, then $v_i^{\mathbf{A}}$ is a $k\|\mathbf{A}\|_\infty \mathcal{L}$-Lipschitz valuation function.*

*Proof.* The statement follows from the fact that $v_i^{\mathbf{A}}(\mathbf{z}) = v_i(\mathbf{Az})$, where $\mathbf{z} \in [0,1]^k$ and $v_i$ is $\mathcal{L}$-Lipschitz. $\qquad\square$

**Lemma 3.** *Given query access to a mechanism $\mathcal{M}$ that is IR and BIC w.r.t. $\hat{\mathcal{F}}_z$, and a sampling algorithm $\mathcal{S}_i$ for each $i \in [n]$, where $\mathcal{S}_i(\mathbf{x}, \delta)$ draws a sample from the conditional distribution of*

$\hat{\mathcal{F}}_{z,i}$ on the $k$-dimensional cube $\times_{j\in[k]}[x_j, x_j + \delta_j)$, we can construct a mechanism $\mathcal{M}_1^\ell$, oblivious to $\hat{\mathcal{F}}$, that is IR and $O(k\|\mathbf{A}\|_\infty \mathcal{L}\delta)$-BIC w.r.t. $\left\lfloor \hat{\mathcal{F}}_z \right\rfloor_{\ell,\delta}$. Furthermore,

$$Rev\left(\mathcal{M}_1^\ell, \left\lfloor \hat{\mathcal{F}}_z \right\rfloor_{\ell,\delta}\right) \geq Rev(\mathcal{M}, \hat{\mathcal{F}}_z) - O(kn\|\mathbf{A}\|_\infty \mathcal{L}\delta).$$

*Proof.* Let $x$ be the allocation rule and $p$ be the payment rule of $\mathcal{M}$. We construct $\mathcal{M}_1^\ell$ as follows. Upon receiving bids $\{\mathbf{w}_i\}_{i\in[n]}$ we first query $\mathcal{S}_i$ for each bidder $i$ to get $\mathbf{w}_i' = \mathcal{S}_i(\mathbf{w}_i, \beta(\mathbf{w}_i))$ where $\beta_j(\mathbf{w}_i) = \delta$ if $w_{i,j} \neq 0$ and $\beta_j(\mathbf{w}_i) = \ell_j$ otherwise. Then, each bidder $i$ gets allocated items $x_i(\mathbf{w}')$ and pays $max\{0, p_i(\mathbf{w}') - k\|\mathbf{A}\|_\infty \mathcal{L}\delta\}$.

We first prove that this mechanism is IR. Note that, by definition of $\mathbf{w}_i$, we have $\|\mathbf{w}_i - \mathbf{w}_i'\|_\infty \leq \delta$ for each $i \in [n]$. Since $\mathcal{M}$ is IR then we know that for all bids $\mathbf{t}_{-i}$ of other bidders, $v_i^{\mathbf{A}}(\mathbf{w}_i', \mathcal{M}(\mathbf{w}_i', \mathbf{t}_{-i})) - p_i(\mathbf{w}_i', \mathbf{t}_{-i}) = u_i(\mathbf{w}_i', \mathcal{M}(\mathbf{w}_i', \mathbf{t}_{-i})) \geq 0$. This gives us the required inequality since,

$$
\begin{aligned}
0 &\leq v_i^{\mathbf{A}}(\mathbf{w}_i', \mathcal{M}(\mathbf{w}_i', \mathbf{t}_{-i})) - p_i(\mathbf{w}_i', \mathbf{t}_{-i}) \\
&= v_i^{\mathbf{A}}(\mathbf{w}_i', \mathcal{M}(\mathbf{w}_i', \mathbf{t}_{-i})) - k\|\mathbf{A}\|_\infty \mathcal{L}\delta - (p_i(\mathbf{w}_i', \mathbf{t}_{-i}) - k\|\mathbf{A}\|_\infty \mathcal{L}\delta) \\
&\leq v_i^{\mathbf{A}}(\mathbf{w}_i, \mathcal{M}(\mathbf{w}_i', \mathbf{t}_{-i})). \hspace{4cm} \text{(via Proposition 1)}
\end{aligned}
$$

We now prove that the mechanism is $O(k\|\mathbf{A}\|_\infty \mathcal{L}\delta)$-BIC. From the point of view of bidder $i$ the types of the other bidders are drawn from $(\hat{\mathcal{F}}_z)_{-i}$ (i.e. $\mathbf{w}_{-i}' \sim (\hat{\mathcal{F}}_z)_{-i}$). From the fact that $\mathcal{M}$ is BIC w.r.t. $\hat{\mathcal{F}}_z$ we have the following:

$$
\begin{aligned}
\mathbb{E}_{\mathbf{w}_{-i}' \sim (\hat{\mathcal{F}}_z)_{-i}} \left[u_i(\mathbf{w}_i, \mathcal{M}(\mathbf{w}_i', \mathbf{w}_{-i}'))\right] &\geq \mathbb{E}_{\mathbf{w}_{-i}' \sim (\hat{\mathcal{F}}_z)_{-i}} \left[u_i(\mathbf{w}_i', \mathcal{M}(\mathbf{w}_i', \mathbf{w}_{-i}'))\right] - k\|\mathbf{A}\|_\infty \mathcal{L}\delta \\
&\geq \max_{\mathbf{x}\in supp(\hat{\mathcal{F}}_{z,i})} \mathbb{E}_{\mathbf{w}_{-i}' \sim (\hat{\mathcal{F}}_z)_{-i}} \left[u_i(\mathbf{w}_i', \mathcal{M}(\mathbf{x}, \mathbf{w}_{-i}'))\right] - k\|\mathbf{A}\|_\infty \mathcal{L}\delta \\
&\geq \max_{\mathbf{x}\in supp(\hat{\mathcal{F}}_{z,i})} \mathbb{E}_{\mathbf{w}_{-i}' \sim (\hat{\mathcal{F}}_z)_{-i}} \left[u_i(\mathbf{w}_i, \mathcal{M}(\mathbf{x}, \mathbf{w}_{-i}'))\right] - 2k\|\mathbf{A}\|_\infty \mathcal{L}\delta.
\end{aligned}
$$
(5)

The first and the third inequalities above follow from Proposition 1. Thus if $i$ could pick the type exactly $\mathbf{w}_i'$ as she pleased, she could not possibly make more than $2k\|\mathbf{A}\|_\infty \mathcal{L}\delta$. However she must pick a $\mathbf{b}_i \in supp\left(\left\lfloor \hat{\mathcal{F}}_{z,i} \right\rfloor_{\ell,\delta}\right)$, which gets rounded to $\mathbf{b}_i' = \mathcal{S}_i(\mathbf{b}_i, \beta(\mathbf{b}_i))$. Specifically, $\mathbf{b}_i' \sim \hat{\mathcal{F}}_{z,i}| \times_{j\in[k]}[b_{i,j}, b_{i,j} + \beta_j(\mathbf{b}_i))$; for notational simplicity, we will simply denote this as $\mathbf{b}_i' \sim \mathcal{S}_i(\mathbf{b}_i, \beta(\mathbf{b}_i))$. Bidder $i$'s utility when she reports bid $b_i \in supp\left(\left\lfloor \hat{\mathcal{F}}_{z,i} \right\rfloor_{\ell,\delta}\right)$ can be bounded using the following observation.

$$
\begin{aligned}
\mathbb{E}_{\mathbf{b}_i' \sim \mathcal{S}_i(\mathbf{b}_i, \beta(\mathbf{b}_i))} \left[u_i(\mathbf{w}_i, \mathcal{M}(\mathbf{b}_i', \mathbf{w}_{-i}'))\right] &= \mathbb{E}_{\mathbf{b}_i' \sim \mathcal{S}_i(\mathbf{b}_i, \beta(\mathbf{b}_i))} \left[\mathbb{E}_{\mathbf{w}_{-i}' \sim (\hat{\mathcal{F}}_z)_{-i}} \left[u_i(\mathbf{w}_i, \mathcal{M}(\mathbf{b}_i', \mathbf{w}_{-i}'))\right]\right] \\
&\leq \max_{\mathbf{x}\in supp(\hat{\mathcal{F}}_{z,i})} \left[\mathbb{E}_{\mathbf{w}_{-i}' \sim (\hat{\mathcal{F}}_z)_{-i}} \left[u_i(\mathbf{w}_i, \mathcal{M}(\mathbf{x}, \mathbf{w}_{-i}'))\right]\right].
\end{aligned}
$$
(6)

On combining inequalities (5) and (6), we get that our new mechanism is $2k\|\mathbf{A}\|_\infty \mathcal{L}\delta$-BIC w.r.t. $\times_{i\in[m]} \left\lfloor \hat{\mathcal{F}}_{z,i} \right\rfloor_{\ell,\delta}$. Finally, note that under truthful bidding this mechanism extracts revenue at least $Rev(\mathcal{M}, \hat{\mathcal{F}}_z) - O(kn\|\mathbf{A}\|_\infty \mathcal{L}\delta)$ since $\mathbf{w}'$ is essentially drawn from $\hat{\mathcal{F}}_z$ and each bidder gets a discount of $k\|\mathbf{A}\|_\infty \mathcal{L}\delta$. $\qquad\square$

**Lemma 4.** *Let* $\left\lfloor \hat{\mathcal{F}}_z \right\rfloor_{\ell,\delta}$ *and* $\lfloor \mathcal{F}_z \rfloor_{\ell,\delta}$ *be distributions such that, for all* $i \in [n]$*, it holds that* (1) $\| \left\lfloor \hat{\mathcal{F}}_{z,i} \right\rfloor_{\ell,\delta} - \lfloor \mathcal{F}_{z,i} \rfloor_{\ell,\delta} \|_{TV} \leq \varepsilon_i^\ell$*, and* (2) $\pi_p \left( \hat{\mathcal{F}}_{z,i}, \mathcal{F}_{z,i} \right) \leq \zeta_p$*. Then, letting* $\rho^\ell := \sum_i \varepsilon_i^\ell$*, given a mechanism* $\mathcal{M}_1^\ell$ *that is IR and* $\eta$*-BIC w.r.t* $\left\lfloor \hat{\mathcal{F}}_z \right\rfloor_{\ell,\delta}$ *we can construct a mechanism* $\mathcal{M}_2^\ell$ *that is IR,* $\left( O(k\mathcal{L}\|\mathbf{A}\|_\infty \rho^\ell + k \left( \zeta_p + \delta \cdot k^{1/p} \right) \|\mathbf{A}\|_\infty \mathcal{L} + \eta), \zeta_p + \delta k^{1/p} \right)$*-BIC w.r.t.* $\lfloor \mathcal{F}_z \rfloor_{\ell,\delta}$*. Furthermore:*

$$Rev \left( \mathcal{M}_2^\ell, \lfloor \mathcal{F}_z \rfloor_{\ell,\delta} \right) \geq Rev \left( \mathcal{M}_1^\ell, \left\lfloor \hat{\mathcal{F}}_z \right\rfloor_{\ell,\delta} \right) - nk\mathcal{L}\|\mathbf{A}\|_\infty \rho^\ell - nk(\zeta_p + \delta k^{1/p})\|\mathbf{A}\|_\infty \mathcal{L}.$$

*Proof.* We will construct $\mathcal{M}_2^\ell$ as follows. For every input bid $\mathbf{w} \in supp \left( \times_{i \in [n]} \lfloor \mathcal{F}_{z,i} \rfloor_{\ell,\delta} \right)$, we first find, for each $i \in [n]$, the closest point in $\ell_p$ norm distance that is in $\left\lfloor \hat{\mathcal{F}}_{z,i} \right\rfloor_{\ell,\delta}$; let $\mathbf{w}_i' = \text{argmin}_{x \in supp\left( \lfloor \hat{\mathcal{F}}_{z,i} \rfloor_{\ell,\delta} \right)} \|\mathbf{w} - \mathbf{x}\|_p$ be this point. Notice here, that we assume that we can calculate the above expression exactly. However, according to Definition 7 we can actually compute $\hat{\mathbf{w}}_i = \text{argmin}_{x \in supp(\hat{\mathcal{F}}_{z,i})} \|\mathbf{w} - \mathbf{x}\|_p$. For the sake of simplicity, we continue the analysis as if we could calculate the desired expression. However, we can set $\mathbf{w}_i' = r^{(\ell,\delta)}(\hat{\mathbf{w}}_i)$ (as defined in Definition 8) and the following proposition implies that we only lose a small factor of $2\delta k^{\frac{1+p}{p}} \|\mathbf{A}\|_\infty \mathcal{L}$ in the BIC guarantee and a $2n\delta k^{\frac{1+p}{p}} \|\mathbf{A}\|_\infty \mathcal{L}$ factor in the revenue guarantee. In the following proposition and for the rest of our analysis we use the notation $\|\mathbf{x} - \mathcal{A}\|_p$ to denote the distance of a vector $\mathbf{x}$ to the closest vector in a set of vectors $\mathcal{A}$, i.e., $\|\mathbf{x} - \mathcal{A}\|_p := \min_{\mathbf{y} \in \mathcal{A}} \|\mathbf{x} - \mathbf{y}\|_p$.

**Proposition 2.** *Let* $\mathcal{B}$ *be a probability distributions supported on* $[0,1]^k$*. Then for any* $\mathbf{x}$ *and* $\mathbf{w} = \text{argmin}_{\mathbf{z} \in supp(\mathcal{B})} \|\mathbf{x} - \mathbf{z}\|_p$ *we have that* $\|\mathbf{x} - r^{(\ell,\delta)}(\mathbf{w})\|_p \leq 2k^{1/p}\delta + \|\mathbf{x} - supp\left( \lfloor \mathcal{B} \rfloor_{\ell,\delta} \right)\|_p$.

*Proof.* For any $\mathbf{x}$ we have that $\|\mathbf{x} - r^{(\ell,\delta)}(\mathbf{x})\|_p \leq k^{1/p}\delta$. Then, by using the triangle inequality for the $\ell_p$-norm and chaining the resulting inequalities, the proposition is implied. $\square$

After finding $\mathbf{w}'$, we then run $\mathcal{M}_1^\ell$ on $\mathbf{w}'$ giving a discount to each bidder, and at the same time making sure that the IR constraint is not violated. Let $x(\cdot)$ and $p(\cdot)$ be the allocation and payment rules of $\mathcal{M}_1^\ell$. For each $i \in [n]$ we do the following: if $\|w_i - w_i'\|_p \leq \left( \zeta_p + \delta \cdot k^{1/p} \right)$, bidder $i$ will receive allocation $x_i(\mathbf{w}')$ and pay $\max\{\hat{p}_i(\mathbf{w}'), 0\}$ where $\hat{p}_i(\mathbf{w}') := p_i(\mathbf{w}') - k \left( \zeta_p + \delta \cdot k^{1/p} \right) \|\mathbf{A}\|_\infty \mathcal{L}$. Otherwise, if $\|w_i - w_i'\|_p > \left( \zeta_p + \delta \cdot k^{1/p} \right)$, she will receive nothing and pays nothing. By construction, and using Proposition 1, mechanism $\mathcal{M}_2^\ell$ is IR.

Next, we show that $\mathcal{M}_2^\ell$ is $\left( O(k\mathcal{L}\|\mathbf{A}\|_\infty \zeta_p + k \left( \zeta_p + \delta \cdot k^{1/p} \right) \|\mathbf{A}\|_\infty \mathcal{L} + \eta), \zeta_p + \delta k^{1/p} \right)$-BIC w.r.t. $\lfloor \mathcal{F}_z \rfloor_{\ell,\delta}$. As a first step, we prove that $\mathcal{M}_2^\ell$ is $O(k \left( \zeta_p + \delta \cdot k^{1/p} \right) \|A\|_\infty \mathcal{L} + \eta)$-BIC w.r.t. $\left\lfloor \hat{\mathcal{F}}_z \right\rfloor_{\ell,\delta}$.

Recall that $\mathcal{M}_1^\ell$ is $\eta$-BIC w.r.t. to $\left\lfloor \hat{\mathcal{F}}_z \right\rfloor_{\ell,\delta}$, i.e., for all bidders $i$, latent types $\mathbf{t}_i, \mathbf{t}_i'$ we have,

$$\mathop{\mathbb{E}}_{\mathbf{b}_{-i} \sim \lfloor (\hat{\mathcal{F}}_z)_{-i} \rfloor_{\ell,\delta}} \left[ u_i(\mathbf{t}_i, \mathcal{M}_1^\ell(\mathbf{t}_i, \mathbf{b}_{-i})) \right] \geq \mathop{\mathbb{E}}_{\mathbf{b}_{-i} \sim \lfloor (\hat{\mathcal{F}}_z)_{-i} \rfloor_{\ell,\delta}} \left[ u_i(\mathbf{t}_i, \mathcal{M}_1^\ell(\mathbf{t}_i', \mathbf{b}_{-i})) \right] - \eta.$$

This further implies that

$$\mathop{\mathbb{E}}_{\mathbf{b}_{-i} \sim \lfloor (\hat{\mathcal{F}}_z)_{-i} \rfloor_{\ell,\delta}} \left[ u_i(\mathbf{t}_i, \mathcal{M}_1^\ell(\mathbf{t}_i, \mathbf{b}_{-i})) + k \left( \zeta_p + \delta \cdot k^{1/p} \right) \|\mathbf{A}\|_\infty \mathcal{L} \right]$$

$$\geq \mathop{\mathbb{E}}_{\mathbf{b}_{-i} \sim \lfloor (\hat{\mathcal{F}}_z)_{-i} \rfloor_{\ell,\delta}} \left[ u_i(\mathbf{t}_i, \mathcal{M}_1^\ell(\mathbf{t}_i', \mathbf{b}_{-i})) + k \left( \zeta_p + \delta \cdot k^{1/p} \right) \|\mathbf{A}\|_\infty \mathcal{L} \right] - \eta$$

$$= \mathop{\mathbb{E}}_{\mathbf{b}_{-i} \sim \lfloor (\hat{\mathcal{F}}_z)_{-i} \rfloor_{\ell,\delta}} \left[ v_i^{\mathbf{A}}(\mathbf{t}_i, x_i(\mathbf{t}_i', \mathbf{b}_{-i})) - p_i(\mathbf{t}_i', \mathbf{b}_{-i}) + k \left( \zeta_p + \delta \cdot k^{1/p} \right) \|\mathbf{A}\|_\infty \mathcal{L} \right] - \eta$$

$$> \mathop{\mathbb{E}}_{\mathbf{b}_{-i} \sim \lfloor (\hat{\mathcal{F}}_z)_{-i} \rfloor_{\ell,\delta}} \left[ v_i^{\mathbf{A}}(\mathbf{t}_i, x_i(\mathbf{t}_i', \mathbf{b}_{-i})) - \max\{\hat{p}_i(\mathbf{t}_i', \mathbf{b}_{-i}), 0\} \right] - \eta. \tag{7}$$

Additionally, using the fact that $x = \max\{x, 0\} + \min\{x, 0\}$ for all $x \in \mathbb{R}$, we can upper bound the left-hand side to obtain,

$$
\mathbb{E}_{\mathbf{b}_{-i} \sim \left\lfloor (\hat{\mathcal{F}}_z)_{-i} \right\rfloor_{\ell,\delta}} \left[ v_i^{\mathbf{A}}(\mathbf{t}_i, x_i(\mathbf{t}_i, \mathbf{b}_{-i})) - p_i(\mathbf{t}_i, \mathbf{b}_{-i}) + k \left( \zeta_p + \delta \cdot k^{1/p} \right) \|\mathbf{A}\|_\infty \mathcal{L} \right]
$$

$$
= \mathbb{E}_{\mathbf{b}_{-i} \sim \left\lfloor (\hat{\mathcal{F}}_z)_{-i} \right\rfloor_{\ell,\delta}} \left[ v_i^{\mathbf{A}}(\mathbf{t}_i, x_i(\mathbf{t}_i, \mathbf{b}_{-i})) - \max\{\hat{p}_i(\mathbf{t}_i', \mathbf{b}_{-i}), 0\} - \min\{\hat{p}_i(\mathbf{t}_i', \mathbf{b}_{-i}), 0\} \right]
$$

$$
\leq \mathbb{E}_{\mathbf{b}_{-i} \sim \left\lfloor (\hat{\mathcal{F}}_z)_{-i} \right\rfloor_{\ell,\delta}} \left[ v_i^{\mathbf{A}}(\mathbf{t}_i, x_i(\mathbf{t}_i, \mathbf{b}_{-i})) - \max\{\hat{p}_i(\mathbf{t}_i', \mathbf{b}_{-i}), 0\} \right] + k \left( \zeta_p + \delta \cdot k^{1/p} \right) \|\mathbf{A}\|_\infty \mathcal{L}.
$$

$$\tag{8}$$

Combining inequalities (7) and (8), gives us the required inequality for the first step, i.e. that $\mathcal{M}_2^\ell$ is $O(k \left( \zeta_p + \delta \cdot k^{1/p} \right) \|A\|_\infty \mathcal{L} + \eta)$-BIC w.r.t. $\left\lfloor \hat{\mathcal{F}}_z \right\rfloor_{\ell,\delta}$.

For the second step, we define, for types $\mathbf{t}_i$, $\mathbf{t}_{-i}$, and $\mathbf{b}_i$, the following auxiliary function $u_i(\mathbf{t}_i \leftarrow \mathbf{b}_i, \mathbf{t}_{-i}) := u_i \left( \mathbf{t}_i, \mathcal{M}_2^\ell(\mathbf{t}_i, \mathbf{t}_{-i}) \right) - u_i \left( \mathbf{t}_i, \mathcal{M}_2^\ell(\mathbf{b}_i, \mathbf{t}_{-i}) \right)$. In other words $u_i(\mathbf{t}_i \leftarrow \mathbf{b}_i, \mathbf{t}_{-i})$ simply represents the difference in utility of $i$ when he reports his true type $\mathbf{t}_i$ as compared to $\mathbf{b}_i$ to the mechanism $\mathcal{M}_2^\ell$. Due to the Lipschitz continuity of valuation functions and the fact that $\mathcal{M}_2^\ell$ is IR, we get that for all choices of $i, \mathbf{t}_i, \mathbf{b}_i, \mathbf{t}_{-i}$ we must have $u_i(\mathbf{t}_i, \mathcal{M}_2^\ell(\mathbf{b}_i, \mathbf{t}_{-i})) \in [-k\mathcal{L}\|\mathbf{A}\|_\infty, k\mathcal{L}\|\mathbf{A}\|_\infty]$. Specifically, this follows from the following two observations: first, $v_i^{\mathbf{A}}(\mathbf{0}, \mathcal{M}_2^\ell(\mathbf{b}_i, \mathbf{t}_{-i})) = 0$, thus due to Lipschitz continuity of $v_i^{\mathbf{A}}$ and the fact that $\mathbf{t}_i \in [0,1]^k$, we have $v_i^{\mathbf{A}}(\mathbf{t}, \mathcal{M}_2^\ell(\mathbf{b}_i, \mathbf{t}_{-i})) \leq k\mathcal{L}\|\mathbf{A}\|_\infty$, and second, that the payments in $\mathcal{M}_2^\ell$ are upper bounded by the maximum utility since $\mathcal{M}_2^\ell$ is IR. Therefore, $-2k\mathcal{L}\|\mathbf{A}\|_\infty \leq u_i(\mathbf{t}_i \leftarrow \mathbf{b}_i, \mathbf{t}_{-i}) \leq 2k\mathcal{L}\|\mathbf{A}\|_\infty$, or equivalently, for all $\mathbf{x}_{-i}, \mathbf{y}_{-i}$ we have that $\mathbb{E}\left[ u_i(\mathbf{t}_i \leftarrow \mathbf{b}_i, \mathbf{x}_{-i}) \right] - \mathbb{E}\left[ u_i(\mathbf{t}_i \leftarrow \mathbf{b}_i, \mathbf{y}_{-i}) \right] \leq 4k\mathcal{L}\|\mathbf{A}\|_\infty \mathbb{1}\{\mathbf{x}_{-i} \neq \mathbf{y}_{-i}\}$, where the expectation is taken over the randomness of the mechanism. This implies that for any coupling $\gamma$ of $\left\lfloor (\hat{\mathcal{F}}_z)_{-i} \right\rfloor_{\ell,\delta}$, $\lfloor (\mathcal{F}_z)_{-i} \rfloor_{\ell,\delta}$ — and in particular for the coupling $\gamma^* = argmin_\gamma \mathbb{E}_{(\mathbf{x}_{-i}, \mathbf{y}_{-i}) \sim \gamma} \left[ \mathbb{1}\{\mathbf{x}_{-i} \neq \mathbf{y}_{-i}\} \right]$ — we have that,

$$
\mathbb{E}_{(\mathbf{x}_{-i}, \mathbf{y}_{-i}) \sim \gamma^*} \left[ u_i(\mathbf{t}_i \leftarrow \mathbf{b}_i, \mathbf{x}_{-i}) - u_i(\mathbf{t}_i \leftarrow \mathbf{b}_i, \mathbf{y}_{-i}) \right] \leq 4k\mathcal{L}\|\mathbf{A}\|_\infty \mathbb{E}_{(\mathbf{x}_{-i}, \mathbf{y}_{-i}) \sim \gamma^*} \left[ \mathbb{1}\{\mathbf{x}_{-i} \neq \mathbf{y}_{-i}\} \right]
$$

$$
\leq 4k\mathcal{L}\|\mathbf{A}\|_\infty \| \left\lfloor (\hat{\mathcal{F}}_z)_{-i} \right\rfloor_{\ell,\delta} - \lfloor (\mathcal{F}_z)_{-i} \rfloor_{\ell,\delta} \|_{TV}
$$

$$
\leq 4k\mathcal{L}\|\mathbf{A}\|_\infty \rho^\ell,
$$

where the final inequality follows from the fact that $\| \left\lfloor (\hat{\mathcal{F}}_z)_{-i} \right\rfloor_{\ell,\delta} - \lfloor (\mathcal{F}_z)_{-i} \rfloor_{\ell,\delta} \|_{TV} \leq \| \left\lfloor \hat{\mathcal{F}}_z \right\rfloor_{\ell,\delta} - \lfloor \mathcal{F}_z \rfloor_{\ell,\delta} \|_{TV} \leq \sum_i \varepsilon_i^\ell = \rho^\ell$. Note that the left hand side can be simplified as $\mathbb{E}_{(\mathbf{x}_{-i}, \mathbf{y}_{-i}) \sim \gamma^*} \left[ u_i(\mathbf{t}_i \leftarrow \mathbf{b}_i, \mathbf{x}_{-i}) - u_i(\mathbf{t}_i \leftarrow \mathbf{b}_i, \mathbf{y}_{-i}) \right] = \mathbb{E}_{\mathbf{x}_{-i} \sim \left\lfloor (\hat{\mathcal{F}}_z)_{-i} \right\rfloor_{\ell,\delta}} \left[ u_i(\mathbf{t}_i \leftarrow \mathbf{b}_i, \mathbf{x}_{-i}) \right] - \mathbb{E}_{\mathbf{y}_{-i} \sim \lfloor (\mathcal{F}_z)_{-i} \rfloor_{\ell,\delta}} \left[ u_i(\mathbf{t}_i \leftarrow \mathbf{b}_i, \mathbf{y}_{-i}) \right]$, and therefore, rearranging give us,

$$
\mathbb{E}_{\mathbf{y}_{-i} \sim \lfloor (\mathcal{F}_z)_{-i} \rfloor_{\ell,\delta}} \left[ u_i \left( \mathbf{t}_i, \mathcal{M}_2^\ell(\mathbf{t}_i, \mathbf{y}_{-i}) \right) \right] - \mathbb{E}_{\mathbf{y}_{-i} \sim \lfloor (\mathcal{F}_z)_{-i} \rfloor_{\ell,\delta}} \left[ u_i \left( \mathbf{t}_i, \mathcal{M}_2^\ell(\mathbf{b}_i, \mathbf{y}_{-i}) \right) \right]
$$

$$
= \mathbb{E}_{\mathbf{y}_{-i} \sim \lfloor (\mathcal{F}_z)_{-i} \rfloor_{\ell,\delta}} \left[ u_i(\mathbf{t}_i \leftarrow \mathbf{b}_i, \mathbf{y}_{-i}) \right]
$$

$$
\geq \mathbb{E}_{\mathbf{x}_{-i} \sim \left\lfloor (\hat{\mathcal{F}}_z)_{-i} \right\rfloor_{\ell,\delta}} \left[ u_i(\mathbf{t}_i \leftarrow \mathbf{b}_i, \mathbf{x}_{-i}) \right] - 4k\mathcal{L}\|\mathbf{A}\|_\infty \rho^\ell. \tag{9}
$$

As the final step of establishing the BIC guarantee of $\mathcal{M}_2^\ell$, we lower bound $\mathbb{E}_{\mathbf{x}_{-i} \sim \left\lfloor (\hat{\mathcal{F}}_z)_{-i} \right\rfloor_{\ell,\delta}} \left[ u_i(\mathbf{t}_i \leftarrow \mathbf{b}_i, \mathbf{x}_{-i}) \right]$. Note that $\mathcal{M}_2^\ell$ maps the reported bids to the closest point in $supp\left( \left\lfloor \hat{\mathcal{F}}_z \right\rfloor_{\ell,\delta} \right)$, and hence, without loss of generality we can assume that when bidders misreport they choose a bid $\mathbf{b}_i \in supp\left( \left\lfloor \hat{\mathcal{F}}_z \right\rfloor_{\ell,\delta} \right)$. Letting $\mathbf{t}_i^* = argmin_{\mathbf{x} \in supp\left( \lfloor \hat{\mathcal{F}}_{z,i} \rfloor_{\ell,\delta} \right)} \|\mathbf{t}_i - \mathbf{x}\|_p$

and $\lambda = \|\mathbf{t}_i - supp\left(\left\lfloor\hat{\mathcal{F}}_z\right\rfloor_{\ell,\delta}\right)\|_p$, we have that,

$$\underset{\mathbf{x}_{-i}\sim\left\lfloor(\hat{\mathcal{F}}_z)_{-i}\right\rfloor_{\ell,\delta}}{\mathbb{E}}\left[u_i(\mathbf{t}_i \leftarrow \mathbf{b}_i, \mathbf{x}_{-i})\right] = \underset{\mathbf{x}_{-i}\sim\left\lfloor(\hat{\mathcal{F}}_z)_{-i}\right\rfloor_{\ell,\delta}}{\mathbb{E}}\left[u_i\left(\mathbf{t}_i, \mathcal{M}_2^\ell(\mathbf{t}_i, \mathbf{x}_{-i})\right) - u_i\left(\mathbf{t}_i, \mathcal{M}_2^\ell(\mathbf{b}_i, \mathbf{x}_{-i})\right)\right]$$

$$= \underset{\mathbf{x}_{-i}\sim\left\lfloor(\hat{\mathcal{F}}_z)_{-i}\right\rfloor_{\ell,\delta}}{\mathbb{E}}\left[u_i\left(\mathbf{t}_i, \mathcal{M}_2^\ell(\mathbf{t}_i^*, \mathbf{x}_{-i})\right) - u_i\left(\mathbf{t}_i, \mathcal{M}_2^\ell(\mathbf{b}_i, \mathbf{x}_{-i})\right)\right]$$

$$\geq \underset{\mathbf{x}_{-i}\sim\left\lfloor(\hat{\mathcal{F}}_z)_{-i}\right\rfloor_{\ell,\delta}}{\mathbb{E}}\left[u_i\left(\mathbf{t}_i^*, \mathcal{M}_2^\ell(\mathbf{t}_i^*, \mathbf{x}_{-i})\right) - u_i\left(\mathbf{t}_i^*, \mathcal{M}_2^\ell(\mathbf{b}_i, \mathbf{x}_{-i})\right)\right] - 2k\mathcal{L}\|\mathbf{A}\|_\infty\lambda.$$

To complete the argument, we use the following two propositions.

**Proposition 3.** *Let $\mathcal{B}$ and $\mathcal{B}'$ be two probability distributions supported on $[0,1]^k$ for which $\pi_p(\mathcal{B}, \mathcal{B}') \leq \varepsilon$. Then it must be true that $\pi_p\left(\lfloor\mathcal{B}\rfloor_{\ell,\delta}, \lfloor\mathcal{B}'\rfloor_{\ell,\delta}\right) \leq \varepsilon + \delta \cdot k^{1/p}$.*

*Proof.* Intuitively, the proposition follows since two probability masses, when rounded, will move at most by an additive factor of $\delta \cdot k^{1/p}$ in the $\ell_p$ norm distance compared to each other. Formally, since $\pi_P(\mathcal{B}, \mathcal{B}') \leq \varepsilon$ from Lemma 1 we know that there exist coupling $\gamma$ of $\mathcal{B}, \mathcal{B}'$ such that $\mathbb{P}_{(x,y)\sim\gamma}\left[\|x - y\|_p > \varepsilon\right] \leq \varepsilon$. From the definition of rounding, we know that $\|r^{(\ell,\delta)}(x) - r^{(\ell,\delta)}(y)\|_p \leq \|x - y + \overrightarrow{\delta}\|_p \leq \|x - y\|_p + \delta\|\overrightarrow{1}\|_p = \|x - y\|_p + \delta \cdot k^{1/p}$. Therefore, for the same coupling $\gamma$ we have that,

$$\underset{(x,y)\sim\gamma}{\mathbb{P}}\left[\|r^{(\ell,\delta)}(x) - r^{(\ell,\delta)}(y)\|_p > \varepsilon + \delta \cdot k^{1/p}\right] \leq \underset{(x,y)\sim\gamma}{\mathbb{P}}\left[\|x - y\|_p + \delta \cdot k^{1/p} > \varepsilon + \delta \cdot k^{1/p}\right]$$

$$\leq \varepsilon \leq \varepsilon + \delta \cdot k^{1/p}.$$

$\square$

**Proposition 4.** *Let $\mathcal{B}$ and $\mathcal{B}'$ be two probability distributions supported on $[0,1]^k$ for which $\pi_p(\mathcal{B}, \mathcal{B}') \leq \varepsilon$. Then it must be true that $\mathbb{P}_{x\sim\mathcal{B}}\left[\|\mathbf{x} - supp(\mathcal{B}')\|_p > \varepsilon\right] \leq \varepsilon$.*

*Proof.* Using Lemma 1 we know that there exist coupling $\gamma$ of $\mathcal{B}, \mathcal{B}'$ such that $\mathbb{P}_{(x,y)\sim\gamma}\left[\|x - y\|_p > \varepsilon\right] \leq \varepsilon$. This directly gives us the required inequality, $\mathbb{P}_{x\sim B}\left[\|x - supp(\mathcal{B}')\|_p > \varepsilon\right] \leq \mathbb{P}_{x\sim\mathcal{B},y\sim\gamma|x}\left[\|x - y\|_p > \varepsilon\right] = \mathbb{P}_{(x,y)\sim\gamma}\left[\|x - y\|_p > \varepsilon\right] \leq \varepsilon$. $\square$

First, as previously shown, our mechanism $\mathcal{M}_2^\ell$ is a $k\left(\zeta_p + \delta \cdot k^{1/p}\right)\|\mathbf{A}\|_\infty\mathcal{L} + \eta$-BIC w.r.t. $\times_{i\in[n]}\left\lfloor\hat{\mathcal{F}}_{z,i}\right\rfloor_{\ell,\delta}$. Therefore, for every $i, \mathbf{t}_i$, and $\mathbf{b}_i$ we have $\mathbb{E}_{\mathbf{t}_{-i}\sim\left\lfloor(\hat{\mathcal{F}}_z)_{-i}\right\rfloor_{\ell,\delta}}\left[u_i(\mathbf{t}_i \leftarrow \mathbf{b}_i, \mathbf{t}_{-i})\right] \geq -k\left(\zeta_p + \delta \cdot k^{1/p}\right)\|\mathbf{A}\|_\infty\mathcal{L} - \eta$. Second, we know that $\pi_p\left(\hat{\mathcal{F}}_{z,i}, \mathcal{F}_{z,i}\right) \leq \zeta_p$, and hence via Proposition 3, we have $\pi_p\left(\left\lfloor\hat{\mathcal{F}}_{z,i}\right\rfloor_{\ell,\delta}, \lfloor\mathcal{F}_{z,i}\rfloor_{\ell,\delta}\right) \leq \zeta_p + \delta \cdot k^{1/p}$. Hence, by invoking Proposition 4, we get that $\lambda = \|\mathbf{t}_i - supp\left(\left\lfloor\hat{\mathcal{F}}_z\right\rfloor_{\ell,\delta}\right)\|_p \leq \zeta_p + \delta \cdot k^{1/p}$ with probability at least $1 - \zeta_p - \delta \cdot k^{1/p}$. Using these two observations we get that,

$$\underset{\mathbf{t}_{-i}\sim\left\lfloor(\hat{\mathcal{F}}_z)_{-i}\right\rfloor_{\ell,\delta}}{\mathbb{E}}\left[u_i(\mathbf{t}_i \leftarrow \mathbf{b}_i, \mathbf{t}_{-i})\right] \geq -k\left(\zeta_p + \delta \cdot k^{1/p}\right)\|\mathbf{A}\|_\infty\mathcal{L} - \eta - 2k\lambda\|\mathbf{A}\|_\infty\mathcal{L}$$

$$\geq -3k\left(\zeta_p + \delta \cdot k^{1/p}\right)\|\mathbf{A}\|_\infty\mathcal{L} - \eta.$$

On combining this with Eq. (9), we get that with probability at least $1 - \zeta_p - \delta \cdot k^{1/p}$,

$$\underset{\mathbf{x}_{-i}\sim\lfloor(\mathcal{F}_z)_{-i}\rfloor_{\ell,\delta}}{\mathbb{E}}\left[u_i\left(\mathbf{t}_i, \mathcal{M}_2^\ell(\mathbf{t}_i, \mathbf{x}_{-i})\right)\right] - \underset{\mathbf{x}_{-i}\sim\lfloor(\mathcal{F}_z)_{-i}\rfloor_{\ell,\delta}}{\mathbb{E}}\left[u_i\left(\mathbf{t}_i, \mathcal{M}_2^\ell(\mathbf{b}_i, \mathbf{x}_{-i})\right)\right]$$

$$\geq -3k\left(\zeta_p + \delta \cdot k^{1/p}\right)\|\mathbf{A}\|_\infty\mathcal{L} - k\mathcal{L}\|\mathbf{A}\|_\infty\rho^\ell - \eta.$$

That is, $\mathcal{M}_2^\ell$ is $\left(O(k\mathcal{L}\|\mathbf{A}\|_\infty\rho^\ell + k\left(\zeta_p + \delta \cdot k^{1/p}\right)\|\mathbf{A}\|_\infty\mathcal{L} + \eta), \zeta_p + \delta k^{1/p}\right)$-BIC w.r.t. $\lfloor\mathcal{F}_z\rfloor_{\ell,\delta}$.

Finally, we prove the revenue guarantee of $\mathcal{M}_2^\ell$. Since our mechanism operates by offering a discount of $k\left(\zeta_p + \delta \cdot k^{1/p}\right)\|\mathbf{A}\|_\infty\mathcal{L}$ to each bidder, we have $Rev\left(\mathcal{M}_2^\ell, \lfloor\hat{\mathcal{F}}_z\rfloor_{\ell,\delta}\right) \geq Rev\left(\mathcal{M}_1^\ell, \lfloor\hat{\mathcal{F}}_z\rfloor_{\ell,\delta}\right) - nk(\zeta_p + \delta k^{1/p})\|\mathbf{A}\|_\infty\mathcal{L}$. Next, we consider the following two cases. Let $\mathbf{t} \sim \lfloor\mathcal{F}_z\rfloor_{\ell,\delta}$ and $\hat{\mathbf{t}} \sim \lfloor\hat{\mathcal{F}}_z\rfloor_{\ell,\delta}$. If $\mathbf{t} = \hat{\mathbf{t}}$, mechanism $\mathcal{M}_2^\ell$ will produce the same revenue. Otherwise, if $\mathbf{t} \neq \hat{\mathbf{t}}$ one instance can extract at most $nk\mathcal{L}\|\mathbf{A}\|_\infty$ more revenue than the other. Since $\|\lfloor\hat{\mathcal{F}}\rfloor_{\ell,\delta} - \lfloor\mathcal{F}\rfloor_{\ell,\delta}\|_{TV} \leq \sum_i \varepsilon_i^\ell = \rho^\ell$, from the characterization of TV distance [LPW09][10] we know that there exists a coupling such that $\mathbf{t} \neq \hat{\mathbf{t}}$ with probability less than $\rho^\ell$. This gives us the desired bound on revenue:

$$Rev\left(\mathcal{M}_2^\ell, \lfloor\mathcal{F}_z\rfloor_{\ell,\delta}\right) \geq Rev\left(\mathcal{M}_2^\ell, \lfloor\hat{\mathcal{F}}_z\rfloor_{\ell,\delta}\right) - nk\mathcal{L}\|\mathbf{A}\|_\infty\rho^\ell$$
$$\geq Rev\left(\mathcal{M}_1^\ell, \lfloor\hat{\mathcal{F}}_z\rfloor_{\ell,\delta}\right) - nk\mathcal{L}\|\mathbf{A}\|_\infty\rho^\ell - nk(\zeta_p + \delta k^{1/p})\|\mathbf{A}\|_\infty\mathcal{L}. \qquad \square$$

**Lemma 5.** *Given a mechanism $\mathcal{M}_2^\ell$ that is IR and $(\eta, \mu)$-BIC w.r.t. $\lfloor\mathcal{F}_z\rfloor_{\ell,\delta}$, we can construct a mechanism $\mathcal{M}^\ell$ that is IR and $(3k\mathcal{L}\|\mathbf{A}\|_\infty\delta + \eta, \mu)$-BIC w.r.t. $\mathcal{F}_z$. Moreover, $Rev\left(\mathcal{M}^\ell, \mathcal{F}_z\right) \geq Rev\left(\mathcal{M}_2^\ell, \lfloor\mathcal{F}_z\rfloor_{\ell,\delta}\right) - nk\mathcal{L}\|\mathbf{A}\|_\infty\delta$.*

*Proof.* Let $x$ be the allocation rule and $p$ be the payment rule of $\mathcal{M}_2^\ell$. The mechanism $\mathcal{M}^\ell$ operates as follows: given the input bid $\mathbf{w}_i$ of each $i$, we first construct $\mathbf{w}_i' = r^{(\ell,\delta)}(\mathbf{w}_i)$, and then we allocate to bidder $i$, the items $x_i(\mathbf{w}')$ and make him pay $\max\{p_i(\mathbf{w}') - k\|\mathbf{A}\|_\infty\mathcal{L}\delta, 0\}$. Note that if $\mathbf{w}_i \sim \mathcal{F}_{z,i}$ then $\mathbf{w}_i' \sim \lfloor\mathcal{F}_{z,i}\rfloor_{\ell,\delta}$.

We first argue that the mechanism is $(3k\mathcal{L}\|\mathbf{A}\|_\infty\delta + \eta, \mu)$-BIC wrt $\mathcal{F}_z$.

$$\underset{\mathbf{t}_{-i}\sim(\mathcal{F}_z)_{-i}}{\mathbb{E}}\left[u_i(\mathbf{w}_i, \mathcal{M}^\ell(\mathbf{w}_i, \mathbf{t}_{-i}))\right] \geq \underset{\mathbf{t}_{-i}'\sim\lfloor(\mathcal{F}_z)_{-i}\rfloor_{\ell,\delta}}{\mathbb{E}}\left[u_i(\mathbf{w}_i, \mathcal{M}_2^\ell(\mathbf{w}_i', \mathbf{t}_{-i}'))\right]$$
$$\geq \underset{\mathbf{t}_{-i}'\sim\lfloor(\mathcal{F}_z)_{-i}\rfloor_{\ell,\delta}}{\mathbb{E}}\left[u_i(\mathbf{w}_i', \mathcal{M}_2^\ell(\mathbf{w}_i', \mathbf{t}_{-i}'))\right] - k\|\mathbf{A}\|_\infty\mathcal{L}\delta.$$

The first inequality follows from the definition of the mechanism $\mathcal{M}_2^\ell$ and the last inequality follows from the fact that $\|\mathbf{w}_i' - \mathbf{w}_i\|_\infty \leq \delta$. Towards completing the proof of the BIC guarantee, we will now lower bound the right-hand side. To this end, note that the mechanism $\mathcal{M}_2^\ell$ is $(\eta, \mu)$-BIC, i.e., with probability at least $1 - \mu$, for any $\mathbf{b}_i \in supp(\mathcal{F}_{z,i})$ we have,

$$\underset{\mathbf{t}_{-i}'\sim\lfloor(\mathcal{F}_z)_{-i}\rfloor_{\ell,\delta}}{\mathbb{E}}\left[u_i(\mathbf{w}_i', \mathcal{M}_2^\ell(\mathbf{w}_i', \mathbf{t}_{-i}'))\right] - k\|\mathbf{A}\|_\infty\mathcal{L}\delta$$
$$\geq \underset{\mathbf{t}_{-i}'\sim\lfloor(\mathcal{F}_z)_{-i}\rfloor_{\ell,\delta}}{\mathbb{E}}\left[u_i(\mathbf{w}_i', \mathcal{M}_2^\ell(r^{(\ell,\delta)}(\mathbf{b}_i), \mathbf{t}_{-i}'))\right] - \eta - k\|\mathbf{A}\|_\infty\mathcal{L}\delta$$
$$\geq \underset{\mathbf{t}_{-i}'\sim\lfloor(\mathcal{F}_z)_{-i}\rfloor_{\ell,\delta}}{\mathbb{E}}\left[u_i(\mathbf{w}_i, \mathcal{M}_2^\ell(r^{(\ell,\delta)}(\mathbf{b}_i), \mathbf{t}_{-i}'))\right] - \eta - 2k\|\mathbf{A}\|_\infty\mathcal{L}\delta$$
$$\geq \underset{\mathbf{t}_{-i}\sim(\mathcal{F}_z)_{-i}}{\mathbb{E}}\left[u_i(\mathbf{w}_i, \mathcal{M}^\ell(r^{(\ell,\delta)}(\mathbf{b}_i), \mathbf{t}_{-i}))\right] - \eta - 3k\|\mathbf{A}\|_\infty\mathcal{L}\delta$$
$$= \underset{\mathbf{t}_{-i}\sim(\mathcal{F}_z)_{-i}}{\mathbb{E}}\left[u_i(\mathbf{w}_i, \mathcal{M}^\ell(\mathbf{b}_i, \mathbf{t}_{-i}))\right] - \eta - 3k\|\mathbf{A}\|_\infty\mathcal{L}\delta,$$

---

[10]That is, $\|X - Y\|_{TV} = \min_\gamma \mathbb{P}_{(X,Y)\sim\gamma}[X \neq Y]$, where $\gamma$ is the minimum over all couplings of $X$ and $Y$.

The second inequality follows from the fact that $\|\mathbf{w}_i' - \mathbf{w}_i\|_\infty \leq \delta$ and the last is due to the definition of the mechanism $\mathcal{M}^\ell$.

We now argue that the mechanism $\mathcal{M}^\ell$ is IR. To establish this, we only need to check instances where $p_i(\mathbf{w}') \neq 0$; if $p_i(\mathbf{w}') = 0$, then the mechanism is trivially IR. Using the fact that $\mathcal{M}_2^\ell$ is IR and $\|\mathbf{w}_i - \mathbf{w}_i'\|_p \leq \delta$, we have that

$$
\begin{aligned}
0 \leq u_i\left(\mathbf{w}_i', \left(\mathcal{M}_2^\ell(\mathbf{w}_i', \mathbf{w}_{-i}')\right)\right) &= v_i^{\mathbf{A}}\left(\mathbf{w}_i', \left(\mathcal{M}_2^\ell(\mathbf{w}_i', \mathbf{w}_{-i}')\right)\right) - p_i(\mathbf{w}_i', \mathbf{w}_{-i}') \\
&\leq v_i^{\mathbf{A}}\left(\mathbf{w}_i, \left(\mathcal{M}_2^\ell(\mathbf{w}_i', \mathbf{w}_{-i}')\right)\right) - p_i(\mathbf{w}_i', \mathbf{w}_{-i}') + k\|\mathbf{A}\|_\infty \mathcal{L}\delta \\
&= u_i\left(\mathbf{w}_i, \left(\mathcal{M}^\ell(\mathbf{w}_i, \mathbf{w}_{-i}')\right)\right).
\end{aligned}
$$

Therefore, $\mathcal{M}^\ell$ is IR. Finally, note that when all bidders bid truthfully $\mathcal{M}^\ell$ extracts the same revenue as $\mathcal{M}_2^\ell$ with a cumulative discount of at most $nk\mathcal{L}\|\mathbf{A}\|_\infty\delta$ for all the bidders, this directly implies the revenue bound stated in the lemma statement. $\qquad\square$

**Lemma 6.** *Let $\mathcal{F}_z$ and $\hat{\mathcal{F}}_z$ be two distributions supported on $\mathbb{R}^k$ such that $\pi_p(\mathcal{F}_z, \hat{\mathcal{F}}_z) \leq \varepsilon$. For any $\delta > 0$, $\mathbb{E}_{\ell \sim \mathcal{U}[0,\delta]^k}\left[\|\lfloor \mathcal{F}_z \rfloor_{\ell,\delta} - \lfloor \hat{\mathcal{F}}_z \rfloor_{\ell,\delta}\|_{TV}\right] \leq \left(1 + \frac{k^{1-1/p}}{\delta}\right)\varepsilon$.*

*Proof.* Using Lemma 1 we know that there exists a coupling $\gamma$ of $\mathcal{F}_z$ and $\hat{\mathcal{F}}_z$ so that $\mathbb{P}_{(x,y)\sim\gamma}[\|x - y\|_p > \varepsilon] \leq \varepsilon$. Thus, we can bound the following probability:

$$
\mathbb{P}_{\ell \sim \mathcal{U}[0,\delta]^k, (x,y)\sim\gamma}\left[r^{(\ell,\delta)}(x) \neq r^{(\ell,\delta)}(y)\right] = \mathbb{P}_{\ell \sim \mathcal{U}[0,\delta]^k, (x,y)\sim\gamma}\left[r^{(\ell,\delta)}(x) \neq r^{(\ell,\delta)}(y) \wedge \|x - y\|_p > \varepsilon\right] +
$$
$$
\mathbb{P}_{\ell \sim \mathcal{U}[0,\delta]^k, (x,y)\sim\gamma}\left[r^{(\ell,\delta)}(x) \neq r^{(\ell,\delta)}(y) \wedge \|x - y\|_p \leq \varepsilon\right].
$$

We can upper bound the first term above by $\mathbb{P}_{(x,y)\sim\gamma}[\|x - y\|_p > \varepsilon] \leq \varepsilon$, and using Bayes rule, the second term can be written as $\mathbb{P}_{\ell \sim \mathcal{U}[0,\delta]^k}\left[r^{(\ell,\delta)}(x) \neq r^{(\ell,\delta)}(y) \mid \|x - y\|_p \leq \varepsilon\right] \cdot \mathbb{P}_{(x,y)\sim\gamma}[\|x - y\|_p \leq \varepsilon] \leq \mathbb{P}_{\ell \sim \mathcal{U}[0,\delta]^k}\left[r^{(\ell,\delta)}(x) \neq r^{(\ell,\delta)}(y) \mid \|x - y\|_p \leq \varepsilon\right]$. Plugging these upper bounds we get,

$$
\begin{aligned}
\mathbb{P}_{\ell \sim \mathcal{U}[0,\delta]^k, (x,y)\sim\gamma}\left[r^{(\ell,\delta)}(x) \neq r^{(\ell,\delta)}(y)\right] &\leq \varepsilon + \mathbb{P}_{\ell \sim \mathcal{U}[0,\delta]^k}\left[r^{(\ell,\delta)}(x) \neq r^{(\ell,\delta)}(y) \mid \|x - y\|_p \leq \varepsilon\right] \\
&\leq \varepsilon + \sum_{i\in[k]} \mathbb{P}_{\ell_i \sim \mathcal{U}[0,\delta]}\left[r_i^{(\ell,\delta)}(x) \neq r_i^{(\ell,\delta)}(y) \mid \|x - y\|_p \leq \varepsilon\right] \\
&\leq \left(1 + \frac{k^{1-1/p}}{\delta}\right)\varepsilon.
\end{aligned}
$$

The final inequality follows from the fact that $\sum_{i\in[k]} \mathbb{P}_{\ell_i \sim \mathcal{U}[0,\delta]}\left[r_i^{(\ell,\delta)}(x) \neq r_i^{(\ell,\delta)}(y)\right] \leq \sum_{i\in[k]} \frac{|x_i - y_i|}{\delta} = \frac{\|x - y\|_1}{\delta} \leq \frac{k^{1-1/p}\|x - y\|_p}{\delta}$, where in the final inequality here we used Holder's inequality. $\qquad\square$

## D   Active learning for regression problems via Randomized Linear Algebra: Details

### D.1   Our query protocol: details for $p = 1$ and $p = 2$

Our query protocol takes as input the archetype matrix $\mathbf{A}$ and a set of sampling probabilities $q_i$, $i \in [d]$, summing up to one. It outputs a sampling (and a rescaling) ma-

trix that can be used to select a small subset of types to query bidders' preferences.

---

**Algorithm 1:** Sampling & Rescaling Algorithm

---

**Input:** $\mathbf{A} \in \mathbb{R}^{d \times k}$, sampling complexity $s_p$, probabilities $q_i > 0$, $i \in [d]$, $\sum_{i=1}^{d} q_i = 1$
**Output:** sampling matrix $\mathbf{S}_p \in \mathbb{R}^{s_p \times d}$, rescaling matrix $\mathbf{D}_p \in \mathbb{R}^{s_p \times s_p}$

Initialize $\mathbf{S}_p$, $\mathbf{D}_p$ to be all-zero matrices;
**for** $t \leftarrow 1$ **to** $s_p$ **do**
    Sample index $j \in [d]$ with respect to the probabilities $q_1 \ldots q_d$;
    $\mathbf{S}_p(t,j) \leftarrow 1$ ;                       `// Set the `$t$`th row of `$\mathbf{S}$` to `$\mathbf{e}_j$
    $\mathbf{D}_p(t,t) \leftarrow \frac{1}{\sqrt{s_p q_j}}$ ;                           `// and rescale`
**end**

---

### D.1.1 The $p = 2$ case

We start with the $p = 2$ case and note that $\ell_2$ regression is the most studied problem in the RLA literature, including the active learning setting. In this case, the sampling probabilities will be the so-called (row) leverage scores of $\mathbf{A}$, which can be computed exactly in $O(dk^2)$ time. Leverage scores can be approximated faster, in time that depends basically on the sparsity of the input matrix and we refer the reader to [M. 11, Woo14, DM16] for details on this very well-studied quantities. The main quality-of-approximation result is captured in the following lemma.

**Lemma 7.** *Let* $\mathbf{A} \in \mathbb{R}^{d \times k}$ *and* $(\mathbf{t} + \boldsymbol{\epsilon}_{nq,2}) \in \mathbb{R}^d$. *Assume that the sampling probabilities* $q_i$ *of Algorithm 1 are the row leverage scores of* $\mathbf{A}$. *Let* $\tilde{\mathbf{z}} \in \mathbb{R}^k$ *be*

$$\tilde{\mathbf{z}} = \arg \min_{\mathbf{z} \in \mathbb{R}^k} \|\mathbf{D}_2 \mathbf{S}_2 \mathbf{A} \mathbf{z} - \mathbf{D}_2 \mathbf{S}_2 (\mathbf{t} + \boldsymbol{\epsilon}_{nq,2})\|_2.$$

*Then, with probability at least* 0.99,

$$\|\mathbf{A}\tilde{\mathbf{z}} - (\mathbf{t} + \boldsymbol{\epsilon}_{nq,p})\|_p \le \gamma_2 \texttt{OPT},$$

*where* $\texttt{OPT} = \min_{\mathbf{z} \in \mathbb{R}^k} \|\mathbf{A}\mathbf{z} - (\mathbf{t} + \boldsymbol{\epsilon}_{nq,2})\|_2$. *Here* $\gamma_2 = 1 + \varepsilon_{\mathbb{Q}}$, *where* $\varepsilon_{\mathbb{Q}} > 0$ *and the query complexity* $s'_2$ *satisfies*

$$s'_2 = O(k \ln k + {}^k/_{\varepsilon_{\mathbb{Q}}}).$$

The proof of the above lemma follows from Lemmas 4 and 5 of [DMMS11], which each hold with probability at least $1 - {}^\delta/_2$, by setting the query complexity to $s_2 = O(k \log k + {}^k/_{\varepsilon_{\mathbb{Q}}})$ and using the leverage scores as sampling probabilities. Applying a union bound over the failure probabilities of the two lemmas concludes the proof.

We now use Theorem 3.3 of [MMWY21] to boost the success probability of the above algorithm. Specifically, we repeat our algorithm $O\left(\ln {}^1/_{\delta'}\right)$ times to derive multiple candidate solutions for any $\delta' \in (0, 1)$. Then, we can use Algorithm 3 of [MMWY21] to select a solution that satisfies a slightly worse accuracy guarantee with high probability. More precisely, our final solution $\tilde{\mathbf{z}}$ will satisfy

$$\|\mathbf{A}\tilde{\mathbf{z}} - (\mathbf{t} + \boldsymbol{\epsilon}_{nq,p})\|_p \le (3 \cdot (1 + \varepsilon_{\mathbb{Q}}) + 2)\texttt{OPT} \tag{10}$$

with probability at least $1 - \delta'$. Fixing $\varepsilon_{\mathbb{Q}} = 0.5$, we get an overall query complexity equal to

$$s_2 = O\left(\ln \left({}^1/_{\delta'}\right) s'_2\right) = O\left(k \ln k \ln \left({}^1/_{\delta'}\right)\right).$$

Eventually, we will need the bound of eqn. (10) to hold for all $n$ bidders via a union bound, so the failure probability must be reduced to $\delta' = {}^\delta/_n$. Recall that $\varepsilon_{\mathbb{Q}} = 0.5$; eqn. (10) becomes

$$\|\mathbf{A}\tilde{\mathbf{z}} - (\mathbf{t} + \boldsymbol{\epsilon}_{nq,p})\|_p \le 6.5 \cdot \texttt{OPT}. \tag{11}$$

The above bound holds with probability at least $1 - \delta' = 1 - {}^\delta/_n$ if

$$s_2 = O\left(k \ln k \ln \left({}^n/_\delta\right)\right),$$

for any $\delta \in (0, 1)$.

### D.1.2 The $p = 1$ case

We now focus on the $p = 1$ case. In this setting, the sampling probabilities will be the Lewis weights, which can be approximated efficiently following the lines of [CP15, MD21]. We do emphasize that approximations to the Lewis weights (or the leverage scores) are sufficient in our setting. We now directly apply Theorem 1.2 of [MD21]). We again need a failure probability that is at most $\delta/n$, since we will need to apply a union bound over all $n$ bidders. In our notation, Theorem 1.2 of [MD21] can be restated as follows.

**Lemma 8.** *Let* $\mathbf{A} \in \mathbb{R}^{d \times k}$ *and* $(\mathbf{t} + \boldsymbol{\epsilon}_{nq,1}) \in \mathbb{R}^d$. *Assume that the sampling probabilities* $q_i$ *of Algorithm 1 are the row Lewis weights ($p = 1$) of* $\mathbf{A}$. *Let* $\tilde{\mathbf{z}} \in \mathbb{R}^k$ *be*

$$\tilde{\mathbf{z}} = \arg\min_{\mathbf{z} \in \mathbb{R}^k} \|\mathbf{D}_1 \mathbf{S}_1 \mathbf{A} \mathbf{z} - \mathbf{D}_1 \mathbf{S}_1 (\mathbf{t} + \boldsymbol{\epsilon}_{nq,1})\|_1.$$

*Then, with probability at least* $1 - (\delta/n)$,

$$\|\mathbf{A}\tilde{\mathbf{z}} - (\mathbf{t} + \boldsymbol{\epsilon}_{nq,1})\|_1 \leq \gamma_1 \mathtt{OPT},$$

*where* $\mathtt{OPT} = \min_{\mathbf{z} \in \mathbb{R}^k} \|\mathbf{A}\mathbf{z} - (\mathbf{t} + \boldsymbol{\epsilon}_{nq,1})\|_1$. *Here* $\gamma_1 = 1 + \varepsilon_{\mathtt{Q}}$, *where* $\varepsilon_{\mathtt{Q}} > 0$ *and the query complexity* $s_1$ *satisfies*

$$s_1 = O\left(k/\varepsilon_{\mathtt{Q}}^2 \ln kn/\varepsilon_{\mathtt{Q}}\delta\right).$$

In our proof of Theorem 1, we will set $\varepsilon_{\mathtt{Q}}$ to 0.5 for simplicity.

### D.2 Our query protocol: details for $p > 2$

In this section, we show how to adapt the results of [MMWY22, MMWY21] in our setting in order to prove Theorem 1). We restate a sequence of results from [MMWY22, MMWY21], focusing on $p > 2$ and using our notation.

**Lemma 9** (Theorem 2.11 in [MMWY21]). *Let* $3 \leq p < \infty$ *be an integer. There exists a randomized algorithm which constructs a sampling matrix* $\mathbf{S}_p \in \mathbb{R}^{s'_p \times d}$ *and a rescaling matrix* $\mathbf{D}_p \in \mathbb{R}^{s'_p \times s'_p}$ *such that, with probability at least* 0.99, *the* $\ell_p$ *subspace embedding property holds:*

$$1/2\|\mathbf{A}\mathbf{x}\|_p \leq \|\mathbf{D}_p \mathbf{S}_p \mathbf{x}\|_p \leq 3/2\|\mathbf{A}\mathbf{x}\|_p, \quad \forall \mathbf{x} \in \mathbb{R}^k.$$

*Using Remark 2.21 [MMWY21], we get*

$$s'_p = O\left(k^{p/2} \ln^3 k\right).$$

Notice that Remark 2.21 [MMWY21] removes the dependency of $s'_p$ on $\ln d$. We also note that we can construct the matrices $\mathbf{D}_p$ and $\mathbf{S}_p$ using Algorithm 1 of [MMWY21]. Then, we solve the sampled $\ell_p$ regression problem $\tilde{\mathbf{z}} = \arg\min_{\mathbf{z} \in \mathbb{R}^k} \|\mathbf{D}_p \mathbf{S}_p \mathbf{A}\mathbf{z} - \mathbf{D}_p \mathbf{S}_p (\mathbf{t} + \boldsymbol{\epsilon}_{nq,p})\|_p$, to get guarantees of the form:

$$\|\mathbf{A}\tilde{\mathbf{z}} - (\mathbf{t} + \boldsymbol{\epsilon}_{nq,p})\|_p \leq \gamma_p \|\mathbf{A}\hat{\mathbf{z}} - (\mathbf{t} + \boldsymbol{\epsilon}_{nq,p})\|_p, \tag{12}$$

for some constant $\gamma_p > 1$ that depends on the choice of $3 \leq p < \infty$. In the above,

$$\hat{\mathbf{z}} = \arg\min_{\mathbf{z} \in \mathbb{R}^k} \|\mathbf{A}\mathbf{z} - (\mathbf{t} + \boldsymbol{\epsilon}_{nq,p})\|_p.$$

**Lemma 10** (Theorem 3.2 in [MMWY21]). *Let* $\mathbf{A} \in \mathbb{R}^{d \times k}$ *and* $(\mathbf{t} + \boldsymbol{\epsilon}_{nq,p}) \in \mathbb{R}^d$. *Let* $3 \leq p < \infty$ *be an integer and*

$$\mathtt{OPT} = \min_{\mathbf{z} \in \mathbb{R}^k} \|\mathbf{A}\mathbf{z} - (\mathbf{t} + \boldsymbol{\epsilon}_{nq,p})\|_p.$$

*If* $\tilde{\mathbf{z}} = \arg\min_{\mathbf{z} \in \mathbb{R}^k} \|\mathbf{D}_p \mathbf{S}_p \mathbf{A}\mathbf{z} - \mathbf{D}_p \mathbf{S}_p (\mathbf{t} + \boldsymbol{\epsilon}_{nq,p})\|_p$, *then, with probability at least* 0.99,

$$\|\mathbf{A}\tilde{\mathbf{z}} - (\mathbf{t} + \boldsymbol{\epsilon}_{nq,p})\|_p \leq 6(200)^{1/p}\mathtt{OPT}.$$

The quantity $(200)^{1/p}$ decreases very fast as $p$ increases. To put things into perspective for $p = 3$, $(200)^{1/p} < 6$. Importantly, to boost the success probability, we can compute $O\left(\ln\left(1/\delta'\right)\right)$ candidate solutions by running the algorithm implied by Lemma 10 multiple times, for any $\delta' \in (0, 1)$. Then, we can use Algorithm 3 of [MMWY21] to select a solution that satisfies a slightly worse accuracy guarantee with high probability. More precisely:

**Lemma 11** (Theorem 3.3 in [MMWY21])**.** *Call Algorithm 3 [MMWY21] with inputs $O\left(\ln\left(1/\delta\right)\right)$ candidate solutions computed by the algorithm of Lemma 10. Then, we get a vector $\tilde{\mathbf{z}} \in \mathbb{R}^k$ such that, with probability at least $1 - \delta'$,*

$$\|\mathbf{A}\tilde{\mathbf{z}} - (\mathbf{t} + \boldsymbol{\epsilon_{nq,p}})\|_p \leq (18(200)^{1/p} + 2)\mathtt{OPT}.$$

Overall, the query complexity for $3 \leq p < \infty$ is

$$s_p = O\left(\ln\left(1/\delta'\right) s_p'\right) = O\left(k^{p/2} \ln^3 k \ln\left(1/\delta'\right)\right).$$

Recall that we will need our bound to hold for all $n$ bidders via a union bound, so the failure probability must be reduced to $\delta' = \delta/n$. Thus, our final sampling complexity $s_p$ is

$$s_p = O\left(k^{p/2} \ln^3 k \ln\left(n/\delta\right)\right),$$

for any $\delta \in (0, 1)$.

### D.3 The proof of Theorem 1

*Proof of Theorem 1.* For notational simplicity, in this proof, we use $\tilde{\mathbf{z}}$ instead of $\mathcal{Q}(\mathbf{t})$. Recall from Definition 5 that we can write (dropping the index $i$ for simplicity):

$$\mathbf{t} = \mathbf{A}\mathbf{z} + \boldsymbol{\epsilon}_{\mathtt{mdl,p}}, \tag{13}$$

where $\|\boldsymbol{\epsilon}_{\mathtt{mdl,p}}\|_p \leq \varepsilon_{\mathtt{mdl},p}$. We now use the active learning protocols of Sections D.1 or D.2 to construct a solution $\tilde{\mathbf{z}} \in \mathbb{R}^k$ such that

$$\|\mathbf{A}\tilde{\mathbf{z}} - (\mathbf{t} + \boldsymbol{\epsilon}_{\mathtt{nq,p}})\|_p \leq \gamma_p \|\mathbf{A}\hat{\mathbf{z}} - (\mathbf{t} + \boldsymbol{\epsilon}_{\mathtt{nq,p}})\|_p, \tag{14}$$

for some constant $\gamma_p > 1$ that depends on the choice of $1 \leq p < \infty$. In the above,

$$\hat{\mathbf{z}} = \arg\min_{\mathbf{z}\in\mathbb{R}^k} \|\mathbf{A}\mathbf{z} - (\mathbf{t} + \boldsymbol{\epsilon}_{\mathtt{nq,p}})\|_p.$$

For $p = 2$, eqn. (11) implies that $\gamma_2 = 6.5$. For $p = 1$, Lemma 8 implies that $\gamma_1 = 1.5$. For $p \geq 3$, Lemma 11 implies that $\gamma_p = 18(200)^{1/p} + 2$.

We now proceed to bound $\|\mathbf{z} - \tilde{\mathbf{z}}\|_p$. To achieve this, we bound the left- and right-hand-sides of eqn. (14). Substitute $\mathbf{t} = \mathbf{A}\mathbf{z} + \boldsymbol{\epsilon}_{\mathtt{mdl,p}}$ to the left hand side to get:

$$
\begin{aligned}
\|\mathbf{A}\tilde{\mathbf{z}} - (\mathbf{t} + \boldsymbol{\epsilon}_{\mathtt{nq,p}})\|_p &= \|\mathbf{A}(\tilde{\mathbf{z}} - \mathbf{z}) - (\boldsymbol{\epsilon}_{\mathtt{nq,p}} + \boldsymbol{\epsilon}_{\mathtt{mdl,p}})\|_p \text{`} \\
&\geq \|\mathbf{A}(\mathbf{z} - \tilde{\mathbf{z}})\|_p - \|\boldsymbol{\epsilon}_{\mathtt{nq,p}} + \boldsymbol{\epsilon}_{\mathtt{mdl,p}}\|_p \\
&\geq \|\mathbf{A}(\mathbf{z} - \tilde{\mathbf{z}})\|_p - (\varepsilon_{\mathtt{nq},p} + \varepsilon_{\mathtt{mdl},p}). \tag{15}
\end{aligned}
$$

The first inequality follows from the reverse triangle inequality and the second from the assumptions on $\boldsymbol{\epsilon}_{\mathtt{nq,p}}$ and $\boldsymbol{\epsilon}_{\mathtt{mdl,p}}$. We now manipulate the right hand side of eqn. (14):

$$
\begin{aligned}
\gamma_p \|\mathbf{A}\hat{\mathbf{z}} - (\mathbf{t} + \boldsymbol{\epsilon}_{\mathtt{nq,p}})\|_p &\leq \gamma_p \|\mathbf{A}\mathbf{z} - (\mathbf{t} + \boldsymbol{\epsilon}_{\mathtt{nq,p}})\|_p \\
&= \gamma_p \| - (\boldsymbol{\epsilon}_{\mathtt{nq,p}} + \boldsymbol{\epsilon}_{\mathtt{mdl,p}})\|_p \\
&\leq \gamma_p (\varepsilon_{\mathtt{nq},p} + \varepsilon_{\mathtt{mdl},p}). \tag{16}
\end{aligned}
$$

The first inequality follows from the fact that $\hat{\mathbf{z}}$ is the optimum solution of eqn. (14) and the second equality follows from eqn. (13). By combining eqns. (14), (15), and (16), we get

$$\|\mathbf{A}(\mathbf{z} - \tilde{\mathbf{z}})\|_p \leq (\gamma_p + 1)(\varepsilon_{\mathtt{nq},p} + \varepsilon_{\mathtt{mdl},p}).$$

Using eqn. (2), we get $\|\mathbf{A}(\mathbf{z} - \tilde{\mathbf{z}})\|_p \geq \sigma_{\min,p}\|\mathbf{z} - \tilde{\mathbf{z}}\|_p$. Let $c_p = \gamma_p + 1$ to conclude:

$$\|\mathbf{z} - \tilde{\mathbf{z}}\|_p \leq \sigma_{\min,p}^{-1}(\mathbf{A}) \cdot c_p(\varepsilon_{\mathtt{nq},p} + \varepsilon_{\mathtt{mdl},p}). \tag{17}$$

Using the $\gamma_p$ values from above, we conclude that for $p = 2$, $c_2 = 7.5$; for $p = 1$, $c_1 = 2.5$; and for $p \geq 3$, $c_p = 18(200)^{1/p} + 3$. (We did not optimize constants, but it is worth noting that reducing $c_p$ below two will need a different approach.) The failure probability of the theorem follows from the failure probability of eqn. (14), which needs to hold for all $n$ bidders with probability at least $1 - \delta$. Applying a union bound and using the failure probabilities presented in Sections D.1 and D.2 concludes the proof of the theorem. $\qquad\square$

## D.4 Improving Theorem 2 of [CD22] and comparisons with our work

We revisit the proof of Theorem 2 of [CD22], using our notation. We present a slightly improved analysis for the $\ell_\infty$ norm case. Let $\mathbf{D}_\infty \mathbf{S}_\infty \mathbf{t}$ and $\mathbf{D}_\infty \mathbf{S}_\infty (\mathbf{t} + \boldsymbol{\epsilon}_{\mathtt{nq},\infty})$ be the bidders' responses to the query protocol $\mathcal{Q}$ with and without the noise of the noisy query model. [CD22] makes the following assumptions, which are completely analogous to our work:

1. $\|\boldsymbol{\epsilon}_{\mathtt{nq},\infty}\|_\infty \leq \varepsilon_{\mathtt{nq},\infty}$ (query noise), and
2. $\|\mathbf{t} - \mathbf{A}\mathbf{z}\|_\infty \leq \varepsilon_{\mathtt{mdl},\infty}$ (model noise).

[CD22] proceeds by solving the induced least squares (instead of $\ell_\infty$) regression problem as follows:

$$\tilde{\mathbf{z}} = \arg\min_{\mathbf{z}\in\mathbb{R}^k} \|\mathbf{D}_\infty \mathbf{S}_\infty \mathbf{A}\mathbf{z} - \mathbf{D}_\infty \mathbf{S}_\infty(\mathbf{t} + \boldsymbol{\epsilon}_{\mathtt{nq},\infty})\|_2$$

$$= \left((\mathbf{D}_\infty \mathbf{S}_\infty \mathbf{A})^T \mathbf{D}_\infty \mathbf{S}_\infty \mathbf{A}\right)^{-1} (\mathbf{D}_\infty \mathbf{S}_\infty \mathbf{A})^T \mathbf{D}_\infty \mathbf{S}_\infty(\mathbf{t} + \boldsymbol{\epsilon}_{\mathtt{nq},\infty}) \qquad (18)$$

$$= (\mathbf{D}_\infty \mathbf{S}_\infty \mathbf{A})^\dagger \mathbf{D}_\infty \mathbf{S}_\infty(\mathbf{t} + \boldsymbol{\epsilon}_{\mathtt{nq},\infty}). \qquad (19)$$

In the above we assume that the inverse of the $k \times k$ matrix $(\mathbf{D}_\infty \mathbf{S}_\infty \mathbf{A})^T \mathbf{D}_\infty \mathbf{S}_\infty \mathbf{A}$ exists, which is also an implicit assumption in [CD22]. Thus, $\mathbf{D}_\infty \mathbf{S}_\infty \mathbf{A} \in \mathbb{R}^{s_\infty \times k}$ has full column rank equal to $k$. However, unlike [CD22], we will use eqn. (19) instead of eqn. (18) in our analysis. This will result in improved bounds with respect to various quantities that arise in the analysis. We now proceed to bound $\|\tilde{\mathbf{z}} - \mathbf{z}\|_\infty$, as follows:

$$\tilde{\mathbf{z}} - \mathbf{z} = (\mathbf{D}_\infty \mathbf{S}_\infty \mathbf{A})^\dagger \left(\mathbf{D}_\infty \mathbf{S}_\infty(\mathbf{t} + \boldsymbol{\epsilon}_{\mathtt{nq},\infty}) - \mathbf{D}_\infty \mathbf{S}_\infty \mathbf{A}\mathbf{z}\right)$$

$$= (\mathbf{D}_\infty \mathbf{S}_\infty \mathbf{A})^\dagger \left(\mathbf{D}_\infty \mathbf{S}_\infty \boldsymbol{\epsilon}_{\mathtt{nq},\infty}\right) + (\mathbf{D}_\infty \mathbf{S}_\infty \mathbf{A})^\dagger \mathbf{D}_\infty \mathbf{S}_\infty(\mathbf{t} - \mathbf{A}\mathbf{z}).$$

The first equality follows by the assumption that $\mathbf{D}_\infty \mathbf{S}_\infty \mathbf{A} \in \mathbb{R}^{s_\infty \times k}$ has full column rank equal to $k$, which implies that $(\mathbf{D}_\infty \mathbf{S}_\infty \mathbf{A})^\dagger \mathbf{D}_\infty \mathbf{S}_\infty \mathbf{A} = \mathbf{I}$. By applying the triangle inequality and using sub-multiplicativity properties of the $\ell_\infty$ norm, we get

$$\|\tilde{\mathbf{z}} - \mathbf{z}\|_\infty \leq \varepsilon_{\mathtt{nq},\infty}\|(\mathbf{D}_\infty \mathbf{S}_\infty \mathbf{A})^\dagger\|_\infty \|\mathbf{D}_\infty\|_\infty \|\mathbf{S}_\infty\|_\infty$$

$$+ \|(\mathbf{D}_\infty \mathbf{S}_\infty \mathbf{A})^\dagger\|_\infty \|\mathbf{D}_\infty\|_\infty \|\mathbf{S}_\infty\|_\infty \|\mathbf{t} - \mathbf{A}\mathbf{z}\|_\infty$$

$$\leq \varepsilon_{\mathtt{nq},\infty}\|(\mathbf{D}_\infty \mathbf{S}_\infty \mathbf{A})^\dagger\|_\infty \|\mathbf{D}_\infty\|_\infty + \varepsilon_{\mathtt{mdl},\infty}\|(\mathbf{D}_\infty \mathbf{S}_\infty \mathbf{A})^\dagger\|_\infty \|\mathbf{D}_\infty\|_\infty$$

$$= (\varepsilon_{\mathtt{nq},\infty} + \varepsilon_{\mathtt{mdl},\infty})\|(\mathbf{D}_\infty \mathbf{S}_\infty \mathbf{A})^\dagger\|_\infty \|\mathbf{D}_\infty\|_\infty$$

$$\leq (\varepsilon_{\mathtt{nq},\infty} + \varepsilon_{\mathtt{mdl},\infty})\sqrt{s_\infty}\|(\mathbf{D}_\infty \mathbf{S}_\infty \mathbf{A})^\dagger\|_2 \|\mathbf{D}_\infty\|_\infty$$

$$\leq \sqrt{s_\infty}(\varepsilon_{\mathtt{nq},\infty} + \varepsilon_{\mathtt{mdl},\infty})\sigma_{\min}^{-1}(\mathbf{D}_\infty \mathbf{S}_\infty \mathbf{A}) \|\mathbf{D}_\infty\|_\infty, \qquad (20)$$

where for a matrix $\mathbf{X} \in \mathbb{R}^{m\times n}$ we have $\|\mathbf{X}\|_\infty = \max_{i\in[m]} \sum_{j=1}^n |\mathbf{X}_{ij}|$ and $\|\mathbf{X}\|_2 = \max_{\|\mathbf{y}\|_2=1} \|\mathbf{X}\mathbf{y}\|_2 = \sigma_{\max}(\mathbf{X})$. In the above we used that fact that $\|\mathbf{S}_\infty\|_\infty = 1$, since $\mathbf{S}_\infty$ is a sampling matrix and the property $\|\mathbf{X}\|_\infty \leq \sqrt{n}\|\mathbf{X}\|_2$ for any matrix $\mathbf{X} \in \mathbb{R}^{m\times n}$. The use of eqn. (19) instead of eqn. (18) improves the bound of [CD22] by avoiding an extra $\sqrt{k}$ term and a quadratic dependency on $\sigma_{\min}(\mathbf{D}_\infty \mathbf{S}_\infty \mathbf{A})$.

The bound of eqn. (20) generalizes the bound of [CD22] to allow for sampling and rescaling of the revealed noisy bidder preferences, as well as the corresponding rows of the archetype matrix $\mathbf{A}$, following the lines of Algorithm 1. It is important to note that in order to get meaningful results using the bound of eqn. (20), we need a lower bound on $\sigma_{\min}(\mathbf{D}_\infty \mathbf{S}_\infty \mathbf{A})$ and an upper bound on $\|\mathbf{D}_\infty\|_\infty$. The approach of [CD22] has no rescaling: it sets $\mathbf{D}_\infty = \mathbf{I}$ and, therefore, $\|\mathbf{D}_\infty\|_\infty = 1$. However, this makes it hard to get non-trivial lower bound for the smallest singular value of the matrix $\mathbf{S}_\infty \mathbf{A}$. This matrix is simply a sample of rows of the matrix $\mathbf{A}$ without any rescaling and its smallest singular value could, in general, be arbitrarily close to zero. As a result, only special cases of the archetype matrix $\mathbf{A}$ were analyzed in [CD22]: for those special cases, the smallest singular value of a sample of rows of the input matrix (without rescaling) can be lower bounded. It is a well-known fact that if one were to form the sampling matrix $\mathbf{S}_\infty$ and the rescaling matrix $\mathbf{D}_\infty$ using Algorithm 1 and the row leverage scores of $\mathbf{A}$ as the sampling probabilities, the smallest singular value of $\mathbf{D}_\infty \mathbf{S}_\infty \mathbf{A}$ would be close to the smallest singular value of $\mathbf{A}$. This, however, cannot be done without rescaling, which would necessitate an upper bound on $\|\mathbf{D}_\infty\|_\infty$. As a concrete example, if all the row leverage scores of $\mathbf{A}$ were equal, as would be the case if $\mathbf{A}$'s columns were a subset of the columns of a

Hadamard matrix, the sampling probabilities would all be equal to $1/d$. In this case, the diagonal rescaling matrix $\mathbf{D}_\infty$ has entries that are propotional to the reciprocal of the sampling probabilities and $\|\mathbf{D}\|_\infty = \sqrt{d/s_\infty}$. This makes the overall error bound depend on the number of bidder types $d$, since $s_\infty$ (the query complexity) is much smaller than $d$. This tension between lower-bounding the smallest singular value of $\mathbf{D}_\infty \mathbf{S}_\infty \mathbf{A}$ and upper bounding the infinity norm of $\mathbf{D}_\infty$ does not seem easy to resolve by following the proposed analysis for arbitrary archetype matrices $\mathbf{A} \in \mathbb{R}^{d \times k}$.

### D.4.1 Discussing the assumptions of [CD22] for the archetype matrix

We conclude this section with a brief discussion of the three settings of Theorem 2 of [CD22] for the archetype matrix. First, for the deterministic structure, the assumption on the archetype matrix $\mathbf{A} \in \mathbb{R}^{d \times k}$ is that it has to contain a square $k \times k$ submatrix $\mathbf{C}$, which is both row- and column-diagonally-dominant. It is unclear whether this is a reasonable assumption in the context of archetype matrices; some connections with the non-negative matrix factorization literature are briefly discussed [DS03]. Importantly, it is unclear what the query protocol $\mathcal{Q}$ will be in this setting, since the query matrix $\mathbf{Q}$ has to satisfy $\mathbf{Q} \cdot \mathbf{A} = \mathbf{C}$ and it is not obvious how to recover $\mathbf{Q}$ if $\mathbf{C}$ is unknown. The other two settings are both probabilistic and draw $k$ archetypes either from a $d$-dimensional Gaussian distribution, or as copies of a $d$-dimensional random vector. In both cases, the archetypes are drawn from distributions that are designed to construct matrices whose columns are approximately orthogonal, at least in expectation. This observation results in large values for $\sigma_{\min}(\mathbf{A})$, and, under additional technical assumptions stated in Theorem 2 and Proposition 1 of [CD22], large values for the smallest singular value of $\mathbf{S}_\infty \mathbf{A}$. Our work directly characterizes the sample complexity of the protocol in terms of a single parameter of the archetype matrix that, intuitively, measures archetype independence with respect to $p$-norms. We provide explicit query protocols that leverage information in the archetype matrix and achieve the promised query complexity. We believe that this is a natural way to connect mechanism design with active learning for regression problems.