# OpenReview forum: "Refined Mechanism Design for Approximately Structured Priors via Active Regression"
_NeurIPS.cc/2023/Conference — NeurIPS 2023 poster_

### Official Review · Reviewer_mHuT · 2023-07-01

**Soundness:** 4 excellent
**Presentation:** 3 good
**Contribution:** 3 good
**Rating:** 7
**Confidence:** 3

**Summary:**

This paper demonstrates that results qualitatively similar to [CD'22] can hold under considerably weaker considers on the latent-factor matrix A, by using randomized linear algebra and a modified mechanism design to circumvent prior restrictions.

EDIT: The rebuttal convinced me of the significance of the work and its contribution, and I have raised my score accordingly.

**Strengths:**

1. This paper substantially weakens the requirements on the design matrix A required by CD22
2. The mechanism design argument based on translating Prokhorov distance into TV distance via rounding seems novel and more broadly useful.
3. This paper makes an interesting connection between randomized linear algebra and mechanism design that is relatively new to the statistical learning community.
4. The comparison to past work is thoroughly explained.
5. The notation table was much appreciated.

**Weaknesses:**

1. It seems that there is quite a lot of notational overheard, and quite a few assumptions. There are many notions of error and approximation floating around and it would be beneficial to try and streamline some of them. For example, having a recapitulation of assumptions in the appendix would be helpful.

2. At times, it feels like the paper is making an incremental contribution to a rather niche setting. Correct me if I'm wrong but it seems that the techniques for RLA are fundamentally standard, even if the notion of approximation is specific to this setting.
3. The authors should try to explain what if any contributions they make to mechanism design / Econ+ML more broadly. One can always imagine defining newer settings with more minutiae. Why is this the "right one"?


In summary, my main concern is whether or not this particular set of assumptions is more "fundamental" than CD'22, and whether the techniques here (esp. in rounding for Mechanism Design), are more broadly useful.


**Questions:**

Are there other Mechanism Design arguments based on Prokhorov's Distance in the literature? Could the results of CD22 be improved or simplified using your proof techniques?

In Defn 4, the authors assume access to "elements of t_i + epsilon_...". Do they mean "entries of"?

**Limitations:**

The setting is somewhat niche. In particular, the RLA model assumes queries of the elements of valuation vectors up to noise. This requires that the principal can elicit somewhat truthful valuations of items from the agent. This assumption is treated non-economically; i.e, the agent is assumed to be at most "epsilon_nq"-strategic. I think the authors should comment on why it is realistic to assume not only query access, but access to (approximately truthful) queries!

---

> ### Author Rebuttal · Authors · 2023-08-03
>
> Thank you for the thoughtful review and questions.
>
> Regarding your comment about the setting being niche, we respectfully disagree. In the last 20 years, internet ad auctions have been extremely influential in advancing auction theory/mechanism design (at least from the CS side), in addition to being a half-a-trillion-dollar industry. From the theoretical side, we understand that designing optimal auctions is difficult (and there are many ways that the literature has quantified what "difficult" means), and also that the complexity/error when designing near-optimal auctions must depend on the (huge) number of items, limiting the applicability of numerous recent results in this domain. This dependence is unavoidable when one assumes richness in the valuation structure of the agents, i.e. when items' values are very different from each other. Removing dependence on the number of items is not a niche issue, but, in our view, a first-order consideration when taking theoretical results to practice. Unfortunately, mechanism design does not yet have a language for quantifying "similarity" between items. One can view our work, and the work of Cai-Daskalakis, as taking the first steps to develop such a language by borrowing technical tools from RLA.
>
> Regarding the truthfulness of queries, our mechanism is approximately truthful. That is, an all-knowing, infinitely powerful agent that lies in the optimal way (lying here means answering queries dishonestly), can only make a little bit more than an agent that doesn't lie. Our (economic) assumption is that if lying optimally can only provide a small benefit, agents will opt for behaving honestly. We find this to be a realistic assumption: if an agent with unbounded power and knowledge can only profit a bit from deviating, a real-world agent won't attempt to spend resources to find a profitable deviation.
>
> We can comment on both of these issues.
>
> Below we answer your questions.
>
> - ``Are there other Mechanism Design arguments based on Prokhorov's Distance in the literature? Could the results of CD22 be improved or simplified using your proof techniques?''
>
> Papers that study the robustness of mechanisms under inaccurate priors often also study the Prokhorov metric, e.g. "Multi-item mechanisms without item-independence: Learnability via robustness" by Brustle et al., or "Posted Pricing and Prophet Inequalities with Inaccurate Priors" by Dutting and Kesselheim. Indeed, we do improve upon CD22 in various ways: we extend the learning component to other norms, we remove assumptions in the mechanism design component, and in the combined component, our analysis distills the parameters that affect the errors, allowing us to push beyond the case of restricted additive valuations.
>
> - ``In Defn 4, the authors assume access to "elements of $t_i$ + ...". Do they mean "entries of"?''
>
> Yes; we will correct this.

---

> > ### Comment · Reviewer_mHuT · 2023-08-14
> > **Thank you for the rebuttal!**
> >
> > I would like to begin by apologizing if I seemed to imply that online auctions is "niche" :). My concern was that, the specific combination of assumptions, criteria and query started to feel somewhat arbitrary.
> >
> > But, upon reading the rebuttal, I am convinced of the novelty of this work. In my first review, I missed the restriction of CD22 to additive valuations, which seems incredibly limiting. Weakening assumptions on the design matrix would make me less enthusiastic, but together with removing additivity I am pleased.  I also appreciate the explanation of the robustness of queries from the perspective of approximate truthfulness.
> >
> > I am raising my score to a 7.

---

> > > ### Author Response · Authors · 2023-08-15
> > >
> > > Thank you for your comment and for updating your score.

---

### Official Review · Reviewer_RniX · 2023-07-04

**Soundness:** 4 excellent
**Presentation:** 3 good
**Contribution:** 3 good
**Rating:** 6
**Confidence:** 3

**Summary:**

The paper studies Bayesian approximate mechanism design in a structured world, where each bidder's preferences can be roughly captured by a "topic model", i.e., the true valuation function is close to a linear combination of a few representative valutions.  Inspired by prior work, the paper follows a two-stage design: the seller first "queries" each bidder and tries to recover the bidder's latent valuation (i.e., coefficients corresponding to the representative valuations)  Then, with the recovered approximate latent valuations, the seller constructs a mechanism by placing blackbox queries to a good mechanism in the latent space.  (Importantly, it is assumed that we already know a good mechanism in the latent space.)  The main result is a design that requires much milder regularity conditions on the representative valuations, which also requires less communication with bidders.  The new design works for metrics induced by $\ell_p$ norms for $p \in [1, \infty)$ whereas prior work focuses on the $\ell_\infty$ norm.

**Strengths:**

The paper is fairly well organized.  In particular, the authors clearly explain their technical contributions in the main paper without giving too much detail, which I appreciate very much.  The paper studies a sensible and well recognized problem and makes solid progress.  Conceptually, I like the "observation" that designing the query scheme is really closely related to central problems in numerical linear algebra, which indeed enables the authors to borrow ideas from that extremely rich body of research.  Technically, the authors give a good overview of their approach, and I'm convinced that there are nontrivial ideas involved.

**Weaknesses:**

The paper is (unavoidably) a bit dense with heavy notation, etc.  I think it would help to use a simple running example to illustrate how things work.  The mechanism is also a bit complex, which is not ideal but perhaps hard to avoid too.

**Questions:**

(Also including detailed comments here)

Line 42, "... often depende on the number of items m, which could be prohibitively large ...": while this claim is largely true, I wonder to what extent this is because of the intrinsic richness of information induced by heterogeneous items.  E.g., if all items are the same then at least the amount of communication needed shouldn't depend on m.  Can you comment on this?

Line 69, conditions on A: do you mean "(i), (ii), *or* (iii)"?

Line 149, $v_i: \mathbb{R}^d \times [0, 1])^m \to \mathbb{R}_+$: so the allocation of a mechanism is one number between [0, 1] for each item?  Is this general enough when bidders have non-additive valuations (e.g., a randomized mechanism may need to specify how items correlate in addition to the marginal probabilities)?  Or do you really mean $\{0, 1\}^m$?

Line 154, "$x: \mathbb{R}_+^{nd} \to [0, 1]^{nm}$": again, why output only the marginal probabilities here?

Line 219, "we don't have bounds of the form ...": this is a bit confusing at this point because Definitions 3 and 5 both talk about bounds precisely of this form.  Can you clarify?

Line 323, Theorem 2: here you do write "$v_i: \mathbb{R}^d \times 2^{[n]}$".  Would be nice to be consistent (also see earlier comments).

---

> ### Author Rebuttal · Authors · 2023-08-03
>
> Thank you for the thoughtful review and questions.
>
> - "Line 42, ... Can you comment on this?"
>
> Indeed, assuming richness of the valuation function is what drives the dependence on $m$ for the results cited. One way to view the current work (and the work of Cai-Daskalakis) is that we take the first steps to move away from such assumptions by borrowing technical tools from the rich field of RLA.
>
> - "Line 69, conditions on A: do you mean "(i), (ii), or (iii)"?"
>
> Yes; we will fix this.
>
> - "Line 149... ", "Line 154: ...", "Line 323: ..."
>
> Thank you for catching this. Indeed, a valuation takes as input a distribution over subsets, not just the marginals. Similarly, the allocation rule outputs a distribution over subsets. We will correct these typos.
>
> - "Line 219, "we don't have bounds of the form ...": this is a bit confusing at this point because Definitions 3 and 5 both talk about bounds precisely of this form. Can you clarify?"
>
> Our assumption is not on the individual types, as Definitions 3 and 5 need, but on the overall distributions. The way this is reconciled in the proofs is by a coupling argument, and specifically Lemma 1 (see lines 336-339). We will clarify this.

---

> > ### Comment · Reviewer_RniX · 2023-08-18
> >
> > Thank you for your response!  I have no further questions.

---

### Official Review · Reviewer_wo8x · 2023-07-06

**Soundness:** 2 fair
**Presentation:** 1 poor
**Contribution:** 2 fair
**Rating:** 5
**Confidence:** 2

**Summary:**

This paper presents an intriguing correlation between mechanism design and randomized linear algebra within regression problems. I find the parallel drawn between the approximation error in active learning for regression issues and $(\epsilon, \delta)$-Bayesian incentive compatibility in mechanism design particularly insightful. The authors further demonstrate, through the delegation of results from the active learning for regression problem, the possibility of creating a higher-dimensional mechanism from a lower-dimensional one.

**Strengths:**

This paper appears to bridge the gap between two previously disconnected problems. This represents a highly intriguing venture in terms of enhancing our understanding of diverse issues and facilitating cross-domain learning. The potential for knowledge transfer and interdisciplinary insights is certainly commendable in this work.

**Weaknesses:**

While this paper tackles potentially insightful ideas, I found it somewhat challenging to follow the authors' explanations. It seems that the essence of their contributions could be further highlighted and articulated. I encourage the authors to strive for greater clarity and more effective communication of their key points to enhance the overall readability and impact of their work.

**Questions:**

I must commend the authors for these novel observations. However, I feel that the paper heavily leans on results already established in the literature, with less emphasis on clarifying the newly introduced correlation. I must confess that I found sections of the paper challenging to grasp. While the contributions of the paper could indeed be noteworthy, they would certainly benefit from improved elucidation. Please consider enhancing the presentation to ensure a more comprehensive understanding for readers.

---

> ### Author Rebuttal · Authors · 2023-08-03
>
> Thank you for the thoughtful review. We appreciate the reviewer's effort to read through our paper and their positive comments regarding knowledge transfer from RLA to mechanism design. We will do our best to improve the presentation of our results, but we would like to highlight that our paper used prior work in innovative ways. We recognize that this is a subjective statement. However, to substantiate it, we would like to mention that we do use prior work on active learning for $\ell_p$ regression problems via randomized linear algebra, but the combination with mechanism design is (again, in our opinion) quite non-trivial: we had to modify the mechanism and analysis of Cai and Daskalakis to operate using $\ell_p$ norms and we had to connect the active learning results with the mechanism design framework, as discussed in the Appendix of our work. We will make every effort to better highlight this novel combination of results in the main paper (instead of the Appendix) in the final version of our paper and include figures (see ``global response'').

---

> > ### Comment · Reviewer_wo8x · 2023-08-20
> >
> > After carefully reading through the feedback from all reviewers, as well as revisiting the paper, I note that a majority of reviewers—4 out of 6—believe the paper successfully bridges two fields and offers significant contributions. In light of this consensus, I am inclined to concur with my esteemed colleagues that the paper merits acceptance. Therefore, I would like to formally recommend the paper for acceptance.

---

### Official Review · Reviewer_Nu2U · 2023-07-11

**Soundness:** 4 excellent
**Presentation:** 3 good
**Contribution:** 3 good
**Rating:** 7
**Confidence:** 4

**Summary:**

In this paper, the authors extend the model proposed by [CD22] to a more general setting and incorporate the techniques from Randomized Linear Algebra (RLA) for active learning literature with revenue-maximizing multi-item mechanism design. Compared with [CD22], this paper extends the infinity norm to any l_p norm when measuring the approximation of query protocol so that the RLA for active learning techniques can be applied to improve the accuracy of the query protocol. In terms of the mechanism design front, the authors relax many restrictive assumptions in [CD22]. This paper seems the first paper utilizing RLA for active learning of regression problems to mechanism design.

**Strengths:**

This paper lies in the intersection between game theory and learning algorithms and it should be of interest to many researchers in this field.

The paper is well-structured and the authors convey the main contributions of the paper in a very clear way, which is easy to understand as a researcher in the EconCS field.

The paper is technically solid and the theoretical results are non-trivial. Even though this paper is regarded as an “incremental” paper built upon [CD22], the technical contribution of this paper is still significant.
Overall I like this paper and recommend the acceptance of this work.


**Weaknesses:**

The novelty of this paper is limited, given it follows the exact same setting in [CD22], but I am not too bothered by this.

It will be better to state Lemma 2 more explicitly, as it is one of the main technical novelties of this paper.


**Questions:**

The authors mention they can also improve the results in [CD22] for the infinity norm setting. Can you provide more discussions in the main context? What is the improvement? How significant compared with [CD22].

How important is the $l_p$ norm metric  in your proof, is all the RLA literature focusing on $l_p$ norm? Is there a way to utilize the techniques there for $l_\infty$ norm. IIUC, the reason you can relax the assumptions (e.g. the relaxed oracle access to the distribution) from [CD22] in the mechanism design front highly depends on $l_p$ norm? Can you provide more discussions regarding this?

---

> ### Author Rebuttal · Authors · 2023-08-03
>
> Thank you for the thoughtful review and questions.
>
> - ``The authors mention they can also improve the results in [CD22] for the infinity norm setting. Can you provide more discussions in the main context? What is the improvement? How significant compared with [CD22].''
>
>
> For the $\ell_\infty$ norm we provide small improvements to the results of Cai and Daskalakis when bounding $\lVert \tilde{z} - z \rVert_{\infty}$. We avoid a quadratic dependence on $\sigma_{min}$ (of a certain matrix) and an extra factor of $\sqrt{k}$. These improvements are discussed in detail in our Appendix, Section D.4. They stem from a better characterization of the $\ell_2$ regression solution that is used as a proxy for the $\ell_\infty$ regression solution and results in better dependencies on the condition number of the archetype matrix and other quantities.
>
> - ``How important is the norm metric in your proof, is all the RLA literature focusing on $\ell_p$ norm? Is there a way to utilize the techniques there for $\ell_\infty$  norm. IIUC, the reason you can relax the assumptions (e.g. the relaxed oracle access to the distribution) from [CD22] in the mechanism design front highly depends on $\ell_p$  norm? Can you provide more discussions regarding this?''
>
> The $\ell_\infty$ norm is an $\ell_p$ norm as well with $p\rightarrow\infty$. There are standard lower-bounds that prove that $\ell_\infty$ regression is impossible, in full generality, in an active learning setting: simply put, one needs to essentially observe all measurements in order to solve $\ell_\infty$ regression, because modifying even a single entry could significantly change the output of the regression problem. This is the fundamental reason underlying the strong assumptions in Cai and Daskalakis: without assuming tremendous structure in the archetype matrix, there is no hope of solving $\ell_\infty$ regression in an active learning setting. And indeed, this is the point of departure for our work: we present mechanisms that work with other, more robust, norms (in the context of active learning at least). Eventually, we present a fully general result for all $\ell_p$ norms, which also shows that as $p\rightarrow \infty$, the active learning setting gets weaker, since the sampling complexity increases. This is precisely the issue with the choice of the $\ell_\infty$ norm in prior work.

---

> > ### Comment · Reviewer_Nu2U · 2023-08-14
> >
> > Thanks for the response. My score remains the same.

---

### Official Review · Reviewer_pEGj · 2023-07-12

**Soundness:** 3 good
**Presentation:** 3 good
**Contribution:** 3 good
**Rating:** 7
**Confidence:** 1

**Summary:**

Based on a model by Cai and Daskalakis [CD22], the authors propose a modular approach to design Bayesian Incentive Compatible mechanisms. In step 1, the algorithm learns the agent's type vector which is a very high dimensional vector. In step 2, given a mechanism based on the low-dimensional type learned in step 1, the author propose a way to robustify the mechanism.

**Strengths:**

1. The interaction between linearized linear algebra and mechanism design is explored, which is very exciting.


**Weaknesses:**

see limitations.

**Questions:**

1. What would be the potential challenges if we want to extend the result to other two-sided market models?
1. A key assumption is that the type distribution admits some form of low-dimensional representation. How could one verify this assumption in realistic settings such as ad auctions for e-commerce?

**Limitations:**

The paper is theoretical and brings out the connection between randomized linear algebra and mechanism design. As a non-expert, I did not have the time to check the proof, but in my opinion, the paper adds valuable insights to the field of mechanism design.

---

> ### Author Rebuttal · Authors · 2023-08-03
>
> Thank you for the thoughtful review and questions. We also thank the reviewer for appreciating the connection between RLA and mechanism, which we believe is an exciting direction for future research.
>
> Below we answer your questions.
>
> - ``What would be the potential challenges if we want to extend the result to other two-sided market models?''
>
> This is a very interesting question for future work. Unfortunately, we lack the expertise in two-sided markets to offer more than an educated guess about concrete technical challenges. Our educated guess is that the main obstacle would be proving a robustification result for two-sided markets. Some very recent papers (e.g. ``On the Optimal Fixed-Price Mechanism in Bilateral Trade'' by Cai and Wu) consider limited information in two-sided markets, but for the machinery in our paper (and the Cai-Daskalakis paper) to extend, much stronger results seem crucial.
>
> - ``A key assumption is that the type distribution admits some form of low-dimensional representation. How could one verify this assumption in realistic settings such as ad auctions for e-commerce?''
>
> We are not aware of any empirical evaluations of this assumption for auctions and e-commerce, and we feel that this would be an interesting topic for future empirical work in this context. Our work was initially motivated by the model of Cai and Daskalakis (whose work also did not consider whether this assumption holds in practice). However, the core idea of using a "low-rank plus noise" model stems from well-established fields like recommender systems and latent semantic indexing. In both of these areas, for over 25 years, user preferences (in the case of recommender systems) or documents (in the case of latent semantic indexing) have been modeled as a linear combination of a small underlying set of prototypes (vectors) plus noise. In the context of latent semantic indexing and recommender systems, such models have served as starting points for many theoretical and experimental approaches, before being replaced and expanded by more complicated and non-linear models. The mechanism design community is still at the stage of understanding the behavior of linear models and our work contributes significantly by employing active learning and randomized linear algebra to fully generalize prior work.

---

### Official Review · Reviewer_NuQU · 2023-07-21

**Soundness:** 2 fair
**Presentation:** 1 poor
**Contribution:** 2 fair
**Rating:** 3
**Confidence:** 2

**Summary:**

This paper studies mechanism design for auctions.  A model of the bidders preferences with a low dimensional latent state is proposed. By interacting with the bidders, the latent state can be estimated with active learning. From here a mechanism design is derived.

Disclaimer: I am not an expert in mechanism design, however I have a background in mathematics, computer science and ML. Unfortunately, I found this paper extremely hard to understand despite significant effort.


**Strengths:**

* The paper brings together results from active learning and mechanism designs.
* The paper refines and strengthens results from prior work.



**Weaknesses:**

I found this paper really challenging to read. The introduction is long, too formal to provide a mtahematical introduction but not formal enough to make sense to give a complete formal picture. After the introduction the setting is formally introduced, creating redundancy and leaving the reader with 6 pages of setup that is extremely hard to understand without additional background knowledge. I am also missing concrete and simple examples to illustrate the setting, or possible a picture/diagram of the seller/bidder interaction.

I cannot evaluate the significance of this work; in my view the work is also not directly related to machine learning (except for invoking some fairly standard results on active learning). Perhaps this paper is better off at a different venue.

To make this paper more accessible, I suggest have a less formal, but more complete introduction (in terms of the background); then the formal setting (possibly simplified), explaining the key ideas of the procedure (not just the results). See my questions below.

I did not comprehend section 3.2 despite a reasonable effort. Despite myself lacking the relevant background, I think the accessibility of this paper needs to be improved before acceptance. The relevance to the ML community is also not clear.

**Questions:**

General questions:
* On a high level, what is the goal and what is optimized? i.e. what are the actions of the seller and the bidders, and what's the interaction/information exchange protocol?
* What is the significance of having n bidders? It seems to me that there is no structure shared between the bidders, so the procedure is applied to each bidder i independently.

line 25: what is "willingness to pay"? is this simply the maximal price the bidder is willing to pay for a subset of items?

line 48/64: Here A \in R^(m times k) but later we have that A \in R^(d times k). This is very confusing. What is the interpretation of the vector Az_i? Are those prices the bidder i is willing to pay?

line 66: "communication-efficient query protocol Q for interacting with each bidder". Most terms here have not been introduced nor can be reasonably inferred from the context.

line 75-79: This paragraph is impossible to understand just from this paper. E.g. u_i, v_i have not been introduced. It is not explained what a transformation of a mechanism is (nor is it clear what the mechanism itself is).

line 101: What is d? Before there was only m, n and k?

line 150: What's a randomized subset? What and how is randomized?

line 154: Some more intuitive explanation of "mechanism" is needed. What are "reported types" what are "allocations of the items and payments?" What's a valuation profile?

line 162-171: What's the relevance of truthful mechanism, and how do we ensure to get a truthful mechanism? Can you explain the equation 166 more intuitively?

line 189: What's a bundle? Previous we had "randomized subsets"? At least make the terminology consistent.

line 223: What was the reason to introduce the index i in the first place?

line 271: What's the protocol "Q"? It seems like the more interesting contributions are left out of the main part of the paper. I would suggest to add some examples to illustrate active learning protocol.

line 332: What's an indirect mechanism?


Minors:

line 24: add range, i.g. "i \in [n]"
display above line 87: What's the range of x?


**Limitations:**

see above

---

> ### Author Rebuttal · Authors · 2023-08-03
>
> Thank you for the thoughtful review and questions. We recognize that it is a difficult paper to comprehend without a background in mechanism design. We respectfully disagree with the reviewer regarding whether this work belongs to NeurIPS: numerous mechanism design papers appear in NeurIPS every year, over the past several years, and one of the topics in the CFP is algorithmic game theory. We also disagree that our work employs fairly standard results in active learning: we do use prior work on active learning for $\ell_p$ regression problems via randomized linear algebra, but the combination with mechanism design is (in our opinion) quite non-trivial.
>
> Below we answer your specific questions.
>
> - ``On a high level, ... exchange protocol?''
>
> The overall problem is, given (query access to) a mechanism for a low-dimensional/structured prior distribution, design a mechanism for the real prior distribution, in a way that is agnostic to the real distribution (but given query/oracle access as described in Theorem 3). The seller's actions are the set of all solutions to this problem; however, the "right" way to think about this is that we take the view of the seller. The bidders simply interact with whatever protocol we provide in a way that their (expected) utility is (approximately) maximized. In terms of the information structure (who knows what), there is an asymmetry: we assume that the bidders know the true prior, while the seller/we do not.
>
>
> - ``What is the significance ... independently.''
>
> Mechanism design for n bidders is very different and much harder than mechanism design for a single bidder, even if the bidders' valuations are mutually independent. See "Multi-Item Auctions Defying Intuition?" by Daskalakis for a collection of interesting observations and results on the very similar phenomenon of selling independent items to a single additive bidder (where, e.g., selling each item optimally is a terrible meechanism)
>
> - "line 25: ...''
>
> Yes.
>
>
> - "line 48/64: ... "
>
> Regarding the difference between line 48 and the main part of the paper, $A \in R^{m \cdot k}$ is the assumption in the Cai-Daskalakis paper. In our work, we can allow matrices where the number of rows is much larger than the number of items. We will further clarify this.
>
> For additive valuations, indeed a type $t_i$ can be thought of as a vector whose $j$-th entry, $t_{i,j}$, is exactly the willingness to pay/valuation of agent $i$ for item $j$. However, a vector/type $t_i$ can have different meanings for different valuation functions. So, we do not make any assumptions about types and only ask that the type of a bidder encodes, in some way, their valuation for the different subsets of items.
>
> - "line 66: ..." and "line 75-79: ..."
>
> We will clarify and elaborate more on those concerns.
>
> - "line 101: ..."
>
> $d$ is the dimension of a type. So, for an additive function, it would be natural to have something like $d=m$, and it's always the case that $d \leq 2^m$ (as we explain in 150-151), but we keep $d$ as a different variable (as opposed to Cai and Daskalakis that have $d=m$) to highlight what affects and what doesn't affect our error bounds.
>
> - "line 150: ..."
>
> A distribution over subsets of items. The valuation function of a bidder is evaluated, typically, on the outcomes of the mechanism. So, if the mechanism allocates to an agent item $1$ with probability $0.5$ and items $2$ and $3$ otherwise, the valuation function takes as input this distribution over bundles and outputs a number (the value of the agent for this distribution over bundles). We will clarify this.
>
> - "line 154: ..."
>
> A mechanism consists of two functions. The allocation function takes as input types and outputs an allocation (a distribution over bundles). The payment function takes as input types and outputs a number for each agent, the amount they owe (or should be paid; payments can be negative). Agents optimize for their own utility, so the types reported to the mechanism might differ from the real types, hence the term "reported types." A valuation profile is the vector of $t = (t_1, \dots, t_n)$ with all the bidders' types. We will further clarify some of these points (e.g., switch "type profile" to "valuations profile" across the paper).
>
> - "line 162-171: ..."
>
> The design of truthful mechanisms is, essentially, the whole point of the field of mechanism design. Here, the mechanism $\hat{M}$ is truthful for the low-dimensional distribution; the goal/point of Thm 2 is to construct an approximately truthful mechanism for the real distribution.
> Equation 166 is saying that, by lying, the probability that an agent gains more than epsilon is smaller than delta.
>
> - "line 223: ... "
>
> For the active learning component, indeed the index $i$ is redundant. However, we thought it would be somehow more confusing to simply drop the index just for 3.1, since $t_i$ is the vector that this component is called on in the overall construction/protocol.
>
> - "line 271: ..."
>
> We describe the details of our protocol in Appendix D. Given the page limit, it was indeed the case that we had to move all main technical contributions to the supplementary material. We will attempt to add such examples, as the reviewer suggests.
>
> - "line 332: ..."
>
> We will further clarify this. The point here is that we defined mechanisms to be "direct," aka take as input types. However, here (Thm 2) the bidders don't directly interact with a mechanism but with the query protocol (that calls a mechanism). Hence the combined process is a game (where truthfulness is a Bayes-Nash equilibrium), not a direct mechanism.

---

> > ### Comment · Reviewer_NuQU · 2023-08-14
> >
> > I'd like to thank the authors for the clarification.
> >
> > I still have the following question:
> > Where exactly is it relevant to have more than one bidder $i$ in the context of the contributions of this work? I.e. I understand that the standard mechanism design allows for multiple bidders, but here, as far as I understand, the bidders are treated individually?
> >
> >
> > I agree now that this work is within the scope of NeurIPS, as also pointed out by the AC.
> >
> > However the concerns about accessibility as well as organization of the paper (i.e. presenting the main algorithm) remain. In particular, I believe that this paper requires a major revision, and needs to be re-evaluated based on the changes. For this reason, I will keep my score at 3 ("Reject").
> >
> > I'd also like to point out that other reviewers seem to share these concerns, despite better final evaluation, in particular w.r.t. presentation (wo8x, RniX, mHuT) and also unclear contributions.

---

> > > ### Author Response · Authors · 2023-08-15
> > >
> > > Thank you for your comment.
> > >
> > > Regarding presentation, part of the complexity comes from needing to use technical language from two, previously disjoint areas. As we already mentioned in the global response, we will do our best to improve the presentation of our work, by including figures and addressing reviewer comments/misunderstandings (so, please give us more concrete feedback if you have any).
> > >
> > > Regarding your question about where it's relevant to have more bidders, our entire reduction/mechanism design component needs to take care of the existence of multiple bidders. This heavily complicates arguing about incentives, since the BIC constraint is a statement about an expectation over other agents' types (drawn from a distribution not known to the mechanism). See, e.g., the proof of Lemmas 3, 4, and 5 in Appendix C.2.

---

### Author Rebuttal · Authors · 2023-08-03

We would like to thank all six reviewers for their reviews and constructive feedback. We recognize that, since our work spans two areas, Randomized Linear Algebra and Mechanism Design, some technical parts of our paper are harder to follow if one does not have a background in both domains. We do believe that our work does open the door to future interactions between these two fundamental areas, that had absolutely no intersection in prior literature. The majority of the reviewers agreed with this assessment. We would also like to mention that NeurIPS has historically accepted numerous papers in both Mechanism Design and Randomized Linear Algebra, therefore we believe that a paper that intersects the two areas very much belongs to this conference.

We commit to making every effort to follow reviewer comments and improve the presentation of our work in order to make it accessible to the broadest possible audience. We will incorporate the valuable suggestions from all reviewers in the final version of this paper. We will also include the attached figure (adjusted to incorporate reviewers' feedback, of course) in the final version of the paper. (Please let us know if the figure is not visible.)

---

### Decision · Program_Chairs · 2023-09-21

**Decision:**

Accept (poster)

**Comment:**

Reviewers generally felt positive about this paper, especially after discussion with the authors. They generally agreed that using ideas from randomized linear algebra to improve upon recent work in mechanism design is a nice contribution. Bridging these two areas is valuable and hopefully will lead to interesting follow up work at their intersection.

Many reviewers felt that the paper was difficult to read, and some recommended rejecting based on this concern. I strongly encourage the authors to incorporate reviewer comments and generally work on improving the presentation for the camera ready version. While a paper spanning two areas can be difficult to write cleanly, it is important to put emphasis on trying to do so as much as possible, if the paper is to have impact and be readable by members of both the RLA and mechanism design communities.